# Neural network cloud top pressure and height for MODIS

Nina Håkansson[1], Claudia Adok[2], Anke Thoss[1], Ronald Scheirer[1], and Sara Hörnquist[1]

[1]Swedish Meteorological and Hydrological Institute (SMHI), Norrköping, Sweden
[2]Regional Cancer Center Western Sweden, Gothenburg, Sweden

**Correspondence:** Nina Håkansson (nina.hakansson@smhi.se)

**Abstract.**

Cloud top height retrieval from imager instruments is important for nowcasting and for satellite climate data records. A neural network approach for cloud top height retrieval from the imager instrument MODIS (Moderate-resolution Imaging Spectro-radiometer) is presented. The neural networks are trained using cloud top layer pressure data from the CALIOP (Cloud-Aerosol Lidar with Orthogonal Polarisation) dataset.

Results are compared with two operational reference algorithms for cloud top height: the MODIS Collection 6 level 2 height product and the cloud top temperature and height algorithm in the 2014 version of the NWC SAF (EUMETSAT (European Organisation for the Exploitation of Meteorological Satellites) Satellite Application Facility for nowcasting and very shortrange forecasting) PPS (Polar Platform System). All three techniques are evaluated using both CALIOP and CPR (CloudSat) (Cloud Profiling Radar for CloudSat (CLOUD SATellite)) height.

Instruments like AVHRR (Advanced Very High Resolution Radiometer) and VIIRS (Visible Infrared Imaging Radiometer Suite) contain fewer channels useful for cloud top height retrievals than MODIS, therefore several different neural networks are investigated to test how infrared channel selection influences retrieval performance. Also a network with only channels available for the AVHRR1 instrument is trained and evaluated. To examine the contribution of different variables, networks with fewer variables are trained. It is shown that variables containing imager information for neighbouring pixels are very important.

The error distributions of the involved cloud top height algorithms are found to be non-Gaussian. Different descriptive statistic measures are presented and it is examplified that bias and SD (standard deviation) can be misleading for non-Gaussian distributions. The median and mode are found to better describe the tendency of the error distributions and IQR (interquartile range) and MAE are found to give the most useful information of the spread of the errors.

For all descriptive statistics presented MAE, IQR, RMSE (root mean square error), SD, mode, median, bias and percentage of absolute errors above 0.25, 0.5, 1 and 2 km the neural network perform better than the reference algorithms both validated with CALIOP and CPR (CloudSat). The neural networks using the brightness temperatures at 11 μm and 12 μm show at least 32 % (or 623 m) lower mean absolute error (MAE) compared to the two operational reference algorithms when validating with CALIOP height. Validation with CPR (CloudSat) height gives at least 25 % (or 430 m) reduction of MAE.

# 1 Introduction

The retrieval of cloud top temperature, pressure and height from imager data from polar orbiting satellites is used both as a vital product in global cloud climatologies (Stubenrauch et al., 2013) and for nowcasting at high latitudes where data from geostationary satellites are either not available or not available in sufficient quality and spatial resolution. Cloud top height products from VIS/IR (visible/infrared) imagers are used in the analysis and early warning of thunderstorm development, for height assignment in aviation forecasts and in data assimilation of atmospheric motion vectors. The cloud top height can serve as input to mesoscale analysis and models for use in nowcasting in general, or as input to other satellite retrievals used in nowcasting (e.g. cloud micro physical properties retrieval, or cloud type retrieval). It is important that climatologists and forecasters have reliable and accurate cloud top height products from recent and past satellite measurements.

There are different traditional techniques to retrieve cloud top height see Hamann et al. (2014) for a presentation of ten cloud top height retrieval algorithms applied to the SEVIRI (Spinning Enhanced Visible Infra-Red Imager). Several algorithms to retrieve cloud top height from polar orbiting satellites are available and used operationally for nowcasting purposes or in cloud climatologies. These include the CTTH (cloud top temperature and height) from the PPS (Polar Platform System) package (Dybbroe et al., 2005), which is also used in the CLARA-A2 (CM SAF (EUMETSAT (European Organisation for the Exploitation of Meteorological Satellites) Satellite Application Facility for Climate Monitoring) cloud, albedo and surface radiation dataset) climate data record (Karlsson et al., 2017), ACHA (Algorithm Working Group (AWG) Cloud Height retrieval Algorithm) used in PATMOS-x (Pathfinder Atmospheres - Extended) (Heidinger et al., 2014), CC4CL (Community Cloud Retrieval for Climate) used in ESA (European Space Agency) Cloud_CCI (Cloud Climate Change Initiative) (Stengel et al., 2017), MODIS (Moderate-resolution Imaging Spectro-radiometer) Collection-6 algorithm (Ackerman et al., 2015) and the ISCCP (International Satellite Cloud Climatology Project) algorithm (Rossow and Schiffer, 1999).

We will use both the MODIS Collection-6 (MODIS-C6) and the version 2014 CTTH from PPS (PPS-v2014) as references to evaluate the performance of neural network based cloud top height retrieval. The MODIS-C6 algorithm is developed for the MODIS instrument. The PPS, delivered by the NWC SAF (EUMETSAT Satellite Application Facility for nowcasting and very shortrange forecasting), is adapted to handle data from instruments AVHRR (Advanced Very High Resolution Radiometer), VIIRS (Visible Infrared Imaging Radiometer Suite) and MODIS.

Artificial neural networks are widely used for non-linear regression problems, see for example Gardner and Dorling (1998), Meng et al. (2007) or Milstein and Blackwell (2016) for neural network applications in atmospheric science. In CC4CL a neural network is used for the cloud detection (Stengel et al., 2017). Artificial neural networks have also been used on MODIS data to retrieve cloud optical depth (Minnis et al., 2016). The COCS (cirrus optical properties derived from CALIOP and SEVIRI algorithm during day and night) algorithm uses artificial neural networks to retrieve cirrus cloud optical thickness and cloud top height for the SEVIRI instrument (Kox et al., 2014). Considering that neural networks in the mentioned examples have successfully derived cloud properties, and that cloud top height retrievals often include fitting of brightness temperatures to temperature profiles, neural network can be expected to retrieve cloud top pressure for MODIS with some skill.

One type of neural network is the multilayer perceptron described in (Gardner and Dorling, 1998) which is a supervised learning technique. If the output for a certain input, when training the multilayer perceptron, is not equal to the target output an error signal is propagated back in the network and the weights of the network are adjusted resulting in a reduced overall error. This algorithm is called the back-propagation algorithm.

In this study we will compare the performance of back-propagation neural network algorithms for retrieving cloud top height (NN-CTTH) with the CTTH algorihm from PPS version 2014 (PPS-v2014) and MODIS Collection 6 (MODIS-C6) algorithm. Several networks will be trained to estimate the contribution of different training variables to the overall result. The networks will be validated using both CALIOP (Cloud-Aerosol Lidar with Orthogonal Polarisation) and CPR (CloudSat) (Cloud Profiling Radar for CloudSat (CLOUD SATellite)) height data.

In section 2 the different datasets used are briefly described and in section 3 the three algorithms are described. Results are presented and discussed in section 4 and final conclusions are found in section 5.

## 2   Instruments and data

For this study we used data from the MODIS instrument on the polar orbiting satellite Aqua in the A-Train, as it is co-located with both CALIPSO (Cloud-Aerosol Lidar and Infrared Pathfinder Satellite Observations) and CloudSat at most latitudes and
has multiple channels useful for cloud top height retrieval.

### 2.1   Aqua - MODIS

The MODIS is a spectro-radiometer with 36 channels covering the solar and thermal spectra. We are using level 1 data from the MODIS instrument on the polar orbiter Aqua. For this study the MYD021km (MODIS Science Data Support Team, 2015a) and MYD03 (MODIS Science Data Support Team, 2015b) for all orbits from 24 dates were used ($1^{st}$ and $14^{th}$ of every month
of 2010). The data were divided into four parts which were used for training, validation during training (used to decide when to quit training), testing under development (used to test different combinations of variables during prototyping) and final validation. The data contains many pixels that are almost identical, because a typical cloud is larger than one pixel. Therefore randomly dividing the data into four datasets is not possible as this would in practice give four identical datasets, which would cause the network to over-train. See Table 1 for distribution of data.

The MODIS Collection-6 climate data records produced by the National Aeronautics and Space Administration (NASA) Earth Observation System was used for comparison. The 1km cloud top height and cloud top pressure from the MYD06_L2-product (Ackerman et al., 2015) for the dates in Table 1 were used.

The satellite zenith angles for MODIS when matched with CALIOP varies between 0.04 and 19.08°and then matched with CPR (CloudSat) it veries between 0.04 and 19.26°.

## 2.2 CALIOP

The CALIOP lidar on the polar orbiting satellite CALIPSO is an active sensor and therefore more sensitive to particle conglomerates with low density than typical imagers. The horizontal pixel resolution is 0.07 km x 0.333 km, this means that when co-locating with MODIS one should remember that CALIOP samples only a small part of each MODIS pixel. The vertical resolution for CALIOP is 30 m. The viewing angle for CALIOP is 3°. The CALIOP 1km Cloud Layer product data were used (for the dates, see Table 1) as the truth to train the networks against, and for validation of the networks. The 1km product was selected because the resolution is closest to the MODIS resolution. For training version 3 was used and for validation version 4, be able to access the improved cloud type information in the feature classification flag in version 4.

## 2.3 CPR (CloudSat)

The CPR (CloudSat) is a radar which derives a vertical profile of cloud water. Its horizontal resolution is 1.4 km x 3.5 km, and its vertical resolution is 0.5 km and the viewing angle is 0.16°. The CPR (CloudSat) product 2B-GEOPROF-R05 (Marchand et al., 2008) was used as an additional source for independent validation of the networks, see Table 1 for selected dates. The validation with CPR (CloudSat) will have a lower percentage of low clouds compared to CALIOP because ground clutter is a problem for space bourne radar instruments.

## 2.4 Other data

Numerical weather prediction (NWP) data are needed as input for the PPS-v2014 and the neural network algorithm. In this study the operational 91-level short-range archived forecasted NWP data from ECMWF (European Centre for Medium-range Weather Forecasting) were used. The analysis times at 00:00 and 12:00 was used and the forecast times (6, 9, 12 and 15 h). Under the period IFS cycles Cycle 35r3, 36r1, 36r3 and 36r4 were operational. Also ice maps (OSI-409 version 1.1) from OSISAF (Satellite Application Facility on Ocean and Sea Ice) were used as input for the PPS cloud mask algorithm.

## 3 Algorithms

### 3.1 PPS-v2014 cloud top temperature and height

The cloud top height algorithm in PPS-v2014, uses two different algorithms for cloud top height retrieval, one for pixels classified as opaque and another for semi-transparent clouds. The reason for having two different algorithms is that the straight forward opaque algorithm can not be used for pixels with optically thin clouds like cirrus or broken cloud fields like cumulus. The signals for these pixels are a mixture of contributions from the cloud itself and underlying clouds and/or the surface.

The algorithm uses a split-window technique to decide whether to apply the opaque or semi-transparent retrieval. All pixels with a difference between the 11 μm and 12 μm brightness temperatures of more than 1.0 K are treated as semi-transparent. This is a slight modification of the PPS version 2014 algorithm where also the clouds classified as non-opaque by cloud type product are considered semi-transparent.

The retrieval for opaque clouds matches the observed brightness temperatures at 11 μm against a temperature profile derived from a short term forecast or (re)analysis of a NWP model, adjusted for atmospheric absorption. The first match, going along the profile from the ground and upwards, gives the cloud top height and pressure. Temperatures colder or warmer than the profile are fitted to, respectively, the coldest or warmest temperature of the profile below tropopause.

The algorithm for semi-transparent pixels uses a histogram method, based on the work of Inoue (1985) and Derrien et al. (1988), which fits a curve to the brightness temperature difference between the 11 μm and 12 μm bands as a function of 11 μm brightness temperatures for all pixels in a segment (32x32 pixels). One parameter of this fitting is the cloud top temperature. The solution is checked for quality (low root mean square error) and sanity (inside physically meaningful interval and not predicted too far from data). The solution is accepted if both tests are passed. The height and pressure are then retrieved from

the temperature, in the same way as for opaque clouds. For more detail about the algorithms see SMHI (2015).

PPS height uses the unit altitude above ground. For all comparisons this is transformed to height above mean sea level, using elevations given in the CPR (CloudSat) or CALIOP datasets.

## 3.2   MODIS Collection 6 Aqua Cloud Top Properties product

In MODIS Collection 6 the $CO_2$-slicing method (described in Menzel et al., 2008) is used to retrieve cloud top pressure using

the 13 and 14 μm channels for ice clouds (as determined from MODIS phase algorithm). For low level clouds the 11 μm channel and the IR-window approach (IRW) with a latitude dependent lapse rate is used over ocean (Baum et al., 2012). Over land the 11 μm temperature is fitted against a 11 μm temperature profile calculated from GDAS (Global Data Assimilation System) temperature, water vapour and ozone profiles and the PFAAST (Pressure-Layer Fast Algorithm for Atmospheric Transmittance) radiative transfer model are used for low clouds (Menzel et al., 2008). For more details about the updates in

Collection 6 see Baum et al. (2012). Cloud pressure is converted to temperature and height using the National Centers for Environmental Prediction Global Data Assimilation System (Baum et al., 2012).

## 3.3   Neural network cloud top temperature and height NN-CTTH

Neural networks are trained using MODIS data co-located with CALIOP data. Nearest neighbour matching was used with the pyresample package in the pyTroll project (Raspaud et al., in press). The Aqua and CALIPSO satellites are both part of the

A-Train and the matched FOV (field of view) are close in time (only 75s apart). The uppermost top layer pressure variable, for both multi- and single-layer clouds, from CALIOP data was used as training truth. Temperature and height for the retrieved cloud top pressure are extracted using NWP-data. Pressure predicted higher than surface pressure are set to surface pressure. For pressures lower than 70 hPa neither height nor temperature values are extracted. The amount of pixels with pressure lower than 70 hPa varies between 0 and 0.05 % for the networks.

### 3.3.1 Neural network variables

To reduce sun-zenith angle dependence and to have the same algorithm for all illumination conditions it was decided to use only infra-red channels to train the neural networks. Several different types of variables were used to train the network. The most basic ones were the NWP temperatures at pressure levels (surface, 950, 850, 700, 500 and 250 hPa). This together with the 11 μm or 12 μm brightness temperature ($B_{11}$ or $B_{12}$) gives the network what is needed to make a radiance fitting to retrieve cloud top pressure for opaque clouds, although with very coarse vertical resolution in the NWP data. For opaque clouds that are geometrically thin, with little or no water vapour above the cloud, the 11 μm and 12 μm brightness temperatures will be the same as the cloud top temperature. If the predicted NWP temperatures are correct the neural network could fit the 11 μm brightness temperature to the NWP temperatures and receive the cloud pressure (similar to what is done in PPS-v2014 and MODIS-C6). For cases without inversions in the temperature profile, the retrieved cloud top pressure should be accurate. The cases with inversions are more difficult to fit correctly, since multiple solutions exist and the temperature inversion might not be accurately captured regarding its strength and height in the NWP data. For semi-transparent clouds the network needs more variables to make a correct retrieval.

To give the network information on opacity of the pixel, brightness temperature difference variables were included ($B_{11} - B_{12}$, $B_{11} - B_{3.7}$, $B_{8.5} - B_{11}$). Texture variables with the standard deviation of brightness temperature, or brightness temperature difference, for 5 x 5 pixels were included. These contain information about whether pixels with large $B_{11} - B_{12}$ are more likely to be semi-transparent or more likely to be fractional or cloud edges.

As described in section 3.1, PPS-v2014 uses $B_{11} - B_{12}$ and $B_{11}$ of the neighbouring pixels to retrieve temperatures for semitransparent clouds. In order to feed the network with some of this information the neighbouring warmest and coldest pixels in $B_{11}$ in a 5 x 5 pixel neighbourhood were identified. Variables using the brightness temperature at these warmest and coldest pixels were calculated, for example the 12 μm brightness temperature for the coldest pixel minus the same for the current pixel: $B_{12}^C - B_{12}$, see Table 2 for more information about what variables were calculated.

The surface pressure was also included, which provides the network with a value for the maximum reasonable pressure. Also the brightness temperature for the $CO_2$ channel at 13.3 μm and the water vapour channels at 6.7 μm and 7.3 μm were included as variables. The $CO_2$ channel at 13.3 μm is used in the $CO_2$-slicing method of MODIS-C6 and should improve the cloud top height retrieval for high clouds.

The instruments AVHRR, VIIRS, MERSI-2 (Medium Resolution Spectral Imager -2) , MetImage (Meteorological Imager) and MODIS all have different selections of IR channels. Most of them have the 11 μm and 12 μm channels. The first AVHRR instrument AVHRR1 had only two IR channels at 11 μm and 3.7 μm and no channel at 12 μm. Networks were trained using combinations of MODIS IR-channels corresponding to the channels available for the other instruments. See Table 3 for specifications of the networks trained. Table 4 gives an overview of what imager channels were used for which network.

To see how much the different variable types contribute to the result, some basic networks were trained using less or no imager data. These are also described in Table 3. Also one network using only NWP data was included as a sanity check. For

this network we expect bad results. However good results for this network would indicate that height information retrieved was already available in the NWP-data.

### 3.3.2 Training

For the training 1.5 million pixels were used, with the distribution 50 % low clouds, 25 % medium level clouds and 25 % high clouds. A higher percentage of low clouds was included because the mean square error (MSE) is often much higher for high clouds. Previous tests showed that less low clouds caused the network to focus too much on predicting the high clouds correctly and showed degraded results for low clouds. For the validation dataset used during training 375000 pixels were randomly selected with the same low/medium/high distribution as for the training data.

The machine learning module Scikit-learn (Pedregosa et al., 2011), the Keras package (Chollet et al., 2015), the Theano (Theano Development Team, 2016) backend and the language Python were used for training the network.

### 3.3.3 Parameters and configurations

During training of the network the MSE was used as the loss function that is minimized during training. The data were standardized by subtracting the mean and dividing with the standard deviation before training.

Choosing the number of hidden neurons and hidden layers of the neural network is also important for the training to be effective. Too few hidden neurons will result in under-fitting. We used two hidden layers with 30 neurons in the first layer and 15 neurons in the second.

The initialization of weights before training the network is important for the neural network to learn faster. There are many different weight initialization methods, for training the networks the glorot uniform weight initialization was used. The activation function used for the hidden layers was the tangent hyperbolic (see Karlik and Olgac, 2011) and for the output layer a linear activation function was used.

To determine the changes in the weights an optimization method is used during the back-propagation algorithm. The optimization method used for the multilayer perceptron is mini-batch stochastic gradient descent which performs mini-batch training. A mini-batch is a sample of observations in the data. Several observations are used to update weights and biases, which is different from the traditional stochastic gradient descent where one observation at a time is used for the updates (Cotter et al., 2011). Having an optimal mini-batch size is important for the training of a neural network because overly large batches can cause the network to take a long time to converge. We used a mini-batch size of 250.

When training the neural network there are different learning parameters that need to be tuned to ensure an effective trainig procedure. During prototyping several different combinations were tested. The learning rate is a parameter that determines the size of change in the weights. A too large learning rate will result in large weight changes and can result in an unstable model (Hu and Weng, 2009). If a learning rate on the other hand is too small the training time of the network will be long. We used a learning rate of 0.01.

The momentum is a parameter which adds a part of the weight change to the current weight change, using momentum can help avoid the network getting trapped in local minima (Gardner and Dorling, 1998). A high value of momentum speeds up

the training of the network. We had a momentum of 0.9. The parameter *learning rate decay*, set to $10^{-6}$, in Keras, is used to decrease the learning rate after each update as the training progresses.

To avoid the neural network from over-fitting (which makes the network extra sensitive to unseen data), a method called *early stopping* was used. In early stopping the validation error is monitored during training to prevent the network from over-
5 fitting. If the validation error is not improved for some (we used 10) epochs training is stopped; this helps to reduce risk of over-fitting. The network for which the validation error was at its lowest is then used. The neural networks were trained for a maximum of 2650 epochs, but the early stopping method caused the training to stop much earlier.

## 4 Results and Discussion

The validation data was matched with CALIOP layer top pressure and layer top altitude or CPR (CloudSat) height using nearest
neighbour matching in the same way as the training data was matched. The CPR (CloudSat) data inludes less clouds as both some very low clouds and some very thin clouds are not detected by the radar. CPR (CloudSat) is included to strengthen the results. There is always a risk that a neural network approach learns also or only the errors of the training truth; however if results are improved also when validated with an independent truth it is made sure that it is not only the errors that are learnt. A cloudy threshold of 30% is used for CPR (CloudSat) to include only strong detections. The coarser vertical resolution for
CPR (CloudSat) of 500m means that MAE is expected to be higher than 250m compared to 15m for CALIOP.

The scatter plots in Figure 4 show how the cloud top pressure retrievals of the neural networks and the reference methods are distributed compared to CALIOP. Figure 3 show the same type of scatter plots for cloud top height with CloudSat as truth. These scatter plots show that all neural networks have similar appearance with most of the data retrieved close to the truth. All methods (NN-CTTH, PPS-v2014 and MODIS-C6) retrieve some heights and pressures that are very far from the true values
of CPR (CloudSat) or CALIOP. It is important to remember that some of these seemingly bad results are due to the different FOV for the MODIS and the CALIOP or CPR (CloudSat) sensors.

Figure 7 compares the NN-AVHRR and PPS-v2014 for one scene. The blue squares for PPS-v2014 (c) are due to the temperature retrieval for 32x32 pixels in one go. We can see that a lot of high clouds are by NN-AVHRR placed higher (pixels that are blue in (c), are white in (a)). For NN-AVHRR in (a) we can see that the large area with low clouds in the lower left
corner gets a consistent cloud top height (the same orange colour everywhere). Note that the NN-AVHRR has a less noisy appearance and has less nodata.

### 4.1 Validation with CALIOP top layer pressure

First we consider the performance of all the trained networks validated with the uppermost CALIOP top layer pressure in terms of mean absolute error (MAE). Results in Table 5 show that both PPS-v2014 and MODIS-C6 have a MAE close to
30 120 hPa. Notice that the network using only the NWP information and no imager channels (NN-NWP) shows high MAE. This was included as a sanity check to see that the neural networks are using mainly the satellite data, and the high MAE for NN-NWP is supporting this. The NN-OPAQUE network using only $B_{12}$ and the basic NWP-data has a 9 hPa improvement in

MAE compared to the reference algorithms. By including the variable $B_{11} - B_{12}$, the MAE improves by an additional 19 hPa because $B_{11} - B_{12}$ contains information about the semi-transparency of the pixel. Adding the NWP variable $Ciwv$, which allows the network to attempt to predict the expected values of $B_{11} - B_{12}$, has a smaller effect of 2 hPa on MAE. However adding all variables containing information on neighboring pixels improves the result by additional 20 hPa. The NN-AVHRR network using 11 μm and 12 μm from MODIS provides an MAE which is reduced by about 50 hPa compared to both from MODIS-C6 and PPS-v2014. Notice also that the scores improve for all categories (low, medium and high) when compared with both PPS-v2014 and MODIS-C6. The inclusion of the neighbouring pixels gives almost 40 % of the improvement. Note that for medium level clouds NN-BASIC-CIWV, without information from neighbouring pixels, has higher MAE compared to PPS-v2014.

Adding more IR channels improves the results further. Adding channel 8.5 μm ($B_{8.5} - B_{11}$, NN-VIIRS) improves MAE by 7 hPa and adding 7.3 μm ($B_{7.3}$, NN-MERSI-2) improves MAE by 5 hPa. Including the other watervapor channel at 6.7 μm ($B_{6.7}$, NN-MetImage-NoCO$_2$) improves MAE only by 1 hPa. The CO$_2$ channel at 13.3 μm ($B_{13.3}$, NN-MetImage) improves the MAE by an additional 6 hPa. The NN-AVHRR1 network trained using 3.7 μm and 11 μm (MAE 76.1 hPa) is a little worse compared to NN-AVHRR (MAE 72.4 hPa). Note that $B_{3.7}$ has a solar component which currently is not treated in any way. If $B_{3.7}$ was corrected for the solar component, by the network or in a preparation step, the results for AVHRR1 might improve. Also NN-AVHRR1 shows better scores for all categories (low, medium, and high) compared to PPS-v2014 and MODIS-C6.

The training with CALIOP using only MODIS from Aqua includes only near NADIR observations with all satellite zenith angles for MODIS below 20°. Figure 1 shows that NN-AVHRR and NN-AVHRR1 networks perform robustly also for higher satellite zenith angles. The NN-VIIRS and NN-MetImage-NoCO$_2$ results deviate for satellite zenith angles larger than 60 degrees. The NN-MERSI-2 results deviate for satellite zenith angles larger than 40 degrees. The NN-MetImage retrieval shows deviations already above 20 degree satellite zenith angles and for satellite zenith angles larger than 40 the retrieval has no predictive skill. Notice that the distribution for MODIS-C6 also depend on the satellite zenith angle (with less high clouds at higher angles). For PPS-v2014 instead less low clouds at higher satellite zenith angles are found. The neural networks (NN-AVHRR, NN-AVHRR1, NN-VIIRS and NN-MetImage-NoCO$_2$) can reproduce the bi-modal cloud top pressure distribution similar to CALIOP, PPS-v2014 deviates from this shape with one peak for mid-level clouds.

## 4.2 Discussion of statistics measures for non-Gaussian error distributions

For pressure we choose a single measure, MAE, to describe the error; however which (and how many) measures are needed to adequately describe the error distribution can be discussed. For a Guassian error distibution the obvious choices are bias and SD (standard deviation) as the Gaussian error distribution is completely determined from bias and SD and all other interesting measures could be derived from bias and SD. Unfortunately the error distributions considered here are non-Gaussian. This is expected, as we know that apart from the errors of the algorithm and the errors due to different FOV we expect the lidar to detect some thin cloud layers not visible to the imager. These thin layers, not detected by the imager, should result in underestimated cloud top heights. In Figure 8 the error distributions for MODIS-C6, PPS-v2014 and NN-AVHRR are shown. The Gaussian error distribution with the same bias and SD are plotted in grey. It is clear that the bias is not at the center (the peak) of the

distribution. The median is not at the center either, but closer to it. For validation with CALIOP we can see the expected negative bias for all algorithms and for all cases we can see that assuming a Gaussian distribution underestimates the amount of small errors.

$$PE_x = \frac{\text{number of absolut errors} > \text{x km}}{\text{number of errors}} \qquad (1)$$

Results compared to CALIOP top layer height and CPR (CloudSat) height are provided for the best performing networks in Table 6 (i.e. NN-OPAQUE, NN-BASIC and NN-BASIC-CIWV was excluded). The skewness show that the distributions are skewed and non-Gaussian. The mode is calculated using the half-range method to robustly etimate the mode from the sample (for more info see Bickel, 2002). The bias should be interpreted with caution. Consider PPS-v2014 compared to CALIOP Table 6 if we add 1465 to all retrievals creating a "corrected" retrieval we would have an error distribution with the same SD

and zero bias but the center (peak) of the distribution would not be closer zero. The $PE_1$ (percentage of absolute errors above 1 km, see Equation 1) for this "corrected" retrieval would increase from 54 % to 73 %! For the user this is clearly not an improvement. The general over estimation of cloud top heights of this "corrected" retrieval would however be detected by the median and the mode which would be further away from zero but now on the positive side. This example illustrates the risk of misinterpretation of the bias for non-Gassian error distributions.

Several different measures of variation are presented in Table 6 MAE, IQR (Interquartile range), SD and RMSE. The measures have different benefits; IQR are robust against outliers and RMSE and SD focuses on the worst retrievals as errors are squared. Considering that it is likely not interesting if useless retrievals with large errors are 10km off or 15km off, in combination with that some large errors are expected due to different FOV and different sensitivities of the instruments, the MAE and IQR provide more interesting measures of variation compared to SD and RMSE. In the example discussed in the previous

section the MAE for the "corrected" retrieval would change only 10 m but the RMSE (Root mean square error) would improve with 356 m indicating a much better algorithm; when in fact it is a degraded algorithm. If the largest errors are considered very important RMSE is prefered over SD for skewed distributions, especially if bias is also presented; as RMSE and bias have a smaller risk to be misinterpreted by the reader as a Gaussian error distribution.

     For low level clouds we have even stronger reasons to expect skewed distributions as there is always a limit (ground) to how

low clouds top heights can be underestimated and Table 7 shows that the skewness is large for low level clouds. The bias for low level clouds is difficult to interpret as it is the combination of the main part of the error distribution located close to zero and the large positve errors (which are to some extent expected due to different FOV). In Figure 2 (f) and (e) the error distributions for MODIS-C6, PPS-v2014 and NN-AVHRR for low level clouds are shown. We can see, in Figure 2, that the NN-AVHRR less often underestimates the cloud top height for low level clouds which partly explains the higher bias for NN-AVHRR.

To examplify the problem with bias and SD for skewed distributions consider PPS-v2014 and NN-AVHRR validated with CPR (CloudSat) in Table 7 and for the argument let us falsely assume a Gaussian error distribution. With this assumption the PPS-v2014 with a 232 m better bias and only 24 m worse SD clearly is the better algorithm. The $PE_2$ and RMSE are very similar between the two algorihtms. However all other measures MAE, IQR, $PE_{0.25}$, $PE_{0.5}$, $PE_1$, median and mode all indicate

that NN-AVHRR is the better algorithm and it is clear in Figure 2 (e) that the NN-AVHRR has the highest and best centered distribution; contrary to what was indicated by the bias and SD given a false asumption of Gaussian error distribution.

One explanation of the low bias for PPS-v2014 validated with CPR (CloudSat) in Table 7 is seen in Figure 2 (e) where the error distribution of PPS-v2014 is shown to be bi-modal; the general small underestimation of cloud top heights compensates for the mode located close to 1.8km. The low bias can also be explained by less low level clouds predicted much too high. The lowest values for $PE_2$, SD and RMSE supports this. If we look at the result for the high clouds (Table 9) we see a large negative tendency for PPS-v2014 (mode and median) and this is also part of the explanation of the small RMSE for PPS-v2014 for low level clouds. If high clouds are generally placed 1.5 km too low; this should improve results for low level clouds mistaken for high. This includes cases where the different FOVs causes the imager to see mostly a high cloud but the lidar and radar see only the part of the FOV with a low cloud. This has a large impact on SD and RMSE as the errors are squared.

Comparing the RMSE, SD for NN-AVHRR and PPS-v2014 for low level clouds in the validation with CPR (CloudSat) also highlights why the RMSE and SD are less useful as measures of variation of the error distribution. The RMSE and SD are very similar between the two algorithms and do not reflect the narrower and better centered error distribution seen for NN-AVHRR for low level clouds in Figure 2 (e). The NN-AVHRR has a larger amount of small errors (see $PE_{0.25}$, $PE_{0.5}$) and only 16 % of the errors are larger than 1 km compared to 29 % for PPS-v2014. But NN-AVHRR has 1 % more absolute errors larger than 2 km and the absolut error for this percent is larger. As the MAE does not square the errors, it indicates instead that the NN-AVHRR has smaller variation of the error distribution. The IQR that does not regard the largest errors at all is more than 500 m better for NN-AVHRR.

The bias of -117 m for NN-AVHRR compared to -1203 m for MODIS-C6 in Table 9 in the validation with CPR (CloudSat) for a Gaussian error distribution would be a large improvement of tendency; however when also considering the mode and the median we can see that the improvement of the tendency is more realistically between 150 to 500 m compared to CPR (CloudSat) and not as large as indicated by the bias.

### 4.3    Validation results with CALIOP and CPR (CloudSat) height

All measures in Table 6 have better values for all neural networks compared to both the reference algorithms and both validation truths. Considering the improvement in all the other measures in Table 6 it is safe to conclude that also the lower bias for the neural networks actually is an improvement. However the mode and median better describe the improvement of tendency and for the mode the worst performing network is just a few meters better than the best mode of the reference algorithms. For the comparison to CALIOP in Table 6 we see that the most measures improve as we add more channels to the neural network. Validated with CPR (CloudSat) the results are not improving for the NN-MetImage-NoCO$_2$ and NN-MetImage. A possible explanation for this can be that some high thin clouds layers are not detected by the radar but the neural network places them higher than the detected CPR (CloudSat) layer below. Thin single layer clouds not detected by the radar are of course not included in the analysis.

In the validation with CALIOP the NN-AVHRR MAE is 623 m lower (corresponding to 32 % reduction of MAE) than MODIS-C6 and 795 m (corresponding to 38 % reduction of MAE) lower than PPS-v2014. The NN-MetImage-NoCO$_2$ has the

best result while performing well at all satellite zenith angles, with a 43 % reduction in MAE when compared to MODIS-C6 and a 48 % reduction when compared to PPS-v2014. The NN-MetImage have even better scores but are not useful for satellite zenith angles exceeding $20°$. In the validation with CPR (CloudSat) the NN-AVHRR shows 430 m lower MAE (corresponding to 25 % reduction of MAE) compared to MODIS-C6 and 482 m (corresponding to 28 % reduction of MAE) compared to PPS-

v2014. The NN-MetImage-NoCO$_2$ shows corresponding to 32 % reduction of MAE compared to MODIS-C6 and correspoding to 34 % reduction of MAE compared to PPS-v2014.

## 4.4   Validation results separated for low, medium and high level clouds

Results for low level clouds (Table 7) show that all distributions are well centered around zero and the median and mode are within 250 m from zero for all algorithms except the mode for PPS-v2014 and NN-MetImageNoCO$_2$ validated with CPR

(CloudSat). The PE$_{0.25}$, PE$_{0.5}$ and PE$_1$ and most useful measures of variation, IQR and MAE, show better values for the neural networks than both reference algorithms as compared to both validation truths. This indicates that the neural networks have a larger amount of good retrievals with small errors. When validation with CALIOP, only 31 % of the absolute errors for NN-AVHRR exceed 0.5 km, compared to 58 % for MODIS-C6 and 47 % for PPS-v2014.

For low level clouds validation with CPR (CloudSat) one needs to keep in mind that some thin cloud layers are not detected

by the radar. This means that the CPR (CloudSat) height does not reflect the true upper most layer for these clouds. Correct cloud top height retrievals for these clouds will give large positive errors in the CPR (CloudSat) validation for low level clouds. This can explain why the PE$_2$ and RMSE for all the neural networks are better than both reference algorithms when validated with CALIOP but when validated with CPR (CloudSat) PPS-v2014 have the best PE$_2$ and RMSE. In Section 4.2 it is discussed why the bias and SD are not very informative for these highly skewed distributions.

Notice that MODIS-C6 has a high MAE (1192 m) for low level clouds when validated with CPR (CloudSat). Also in the CALIOP validation MODIS-C6 has the highest MAE, IQR, RMSE, PE$_{0.25}$, PE$_{0.5}$, PE$_1$ and PE$_2$ for low level clouds. When checking the MAE per month we found that scores for MODIS-C6 for low clouds were worst for December (at the same time the scores for high clouds were best in December). There turned out to be a bug in the algorithm for low marine cloud top height (Richard Frey, MODIS Team, 2017 pers. comm.) which likely affected the results and the bug has been corrected in

Version 6.1. However overall validation scores for MODIS-C6 were not affected by the bug (Steve Ackerman, MODIS Team, 2017 pers. comm.).

For medium level clouds (see Table 8) the neural networks have better measures for MAE, IQR, RMSE, SD, PE$_1$, PE$_2$ compared to both reference algorithms when validated both with CALIOP and CPR (CloudSat). For the validation with CPR (Cloudsat) the neural network also have the best PE$_{0.25}$, PE$_{0.5}$, median and bias. In the validation with CALIOP we can see

that also PPS-v2014 has good values for PE$_{0.25}$, PE$_{0.5}$, median and the bias even better than some of the neural networks. This is also seen in Figure 2 (d) where we can see that PPS-v2014 has a well centered and high peak for the error distribution, but a larger amount of underestimated cloud top heights compared to NN-AVHRR. All algorithms report good values for the mode within 300 m from zero for medium level clouds.

For high clouds, in Figure 2, we can see that the NN-AVHRR has less clouds predicted too low, especially compared to PPS-v2014. In the validation with CALIOP (Table 9) the neural networks perform better than the two reference algorithms. For the high clouds validation with CPR (CloudSat) MODIS-C6 has the highest peak (Figure 2), but also a bi-modal error distribution with another peak close to -6 km. This explains why the overall MAE (Table 9) for high clouds is better for the NN-AVHRR. The higher peak for MODIS-C6 for validation with CPR (CloudSat) is also reflected in a good IQR, $PE_{0.5}$, $PE_1$ and mode inline with the neural networks.

The median and mode for high level clouds for most neural networks are positive when compared to CPR (CloudSat) but negative when valdiated with CALIOP. This supports the idea that some high thin clouds, or upper part of clouds, are not detected by the radar but by the lidar and the imager. The median for the neural networks for high level clouds are increasing for neural networks with more variables. This suggests that the extra channels help the neural networks to detect the very thin clouds detected by CALIOP. The medians for the validation with CPR (CloudSat) are also increasing, becoming more positive, and this can be explained by some very thin cloud layers not detected by CPR (CloudSat).

In Table 9 we can also note that SD for the PPS-v2014 validated with CPR (CloudSat) is in line with SD for the neural networks. This in combination with the large negative values on mode and median, and the high MAE and quite good IQR indicates that PPS-v2014 systematically underestimates the cloud top height for high-level clouds.

### 4.5  Validation with CALIOP separated for different cloudtypes

In Table 10, the MAE, median and $PE_{0.5}$ are shown for the different cloud types from the CALIOP feature classification flag. We can see that the MAE and $PE_{0.5}$ for all the neural networks is better than both reference algorithms, except that PPS-v2014 also has a low MAE and $PE_{0.5}$ for *opaque altostratus*. Large improvements in MAE are seen for the *altocumulus transparent transparent cirrus* and *deep convective (opaque)* classes. For $PE_{0.5}$ the largest improvements is seen for the four low cloud classes and the *deep convective (opaque)* class for which the neural networks have at least 12 % less errors above 0.5 km compared to both reference algorithms.

All algorithms have medians closer to zero than 250 m for the classes *low overcast (transparent)* and *transition stratocumulus*. For the *low overcast (opaque)* and *low broken cumulus* the neural networks and PPS-v2014 show good values for the median. For the classes *altocumulus transparent*, *transparent cirrus* and *deep convective (opaque)* clouds the neural network show medians at least 450 m closer to zero than both reference algorithms. For the *opaque altostratus* class the median of the reference algorithms is better than the neural networks. PPS-v2014 also have a MAE and $PE_{0.5}$ that is better than NN-AVHRR and NN-AVHRR1 for the *opaque altostratus* class. The good performance of PPS-v2014 for *opaque altostratus* are also reflected in Figure 2 (d) where PPS-v2014 have the highest peak.

It is most difficult for all algorithms to correctly retrieve cloud top height for the largest class *cirrus (transparent)*. If we compare NN-MetImage with PPS-v2014 for the *cirrus (transparent)* class we see that MAE is improved with 2.4 km, the median with 3 km and 21 % less absolute errors are larger than 500 m.

## 4.6   Geographical aspects of the NN-CTTH performance

To show how performance varies between surfaces and different parts of the globe, the MAE in meters compared to CALIOP are calculated on a Fibonacci grid (constructed using the method described in González, 2009) with a grid evenly spread out on the globe approximately 250 km apart. All observations are matched to the closest grid point and results are plotted in Figure 5. We can see that all algorithms have problems with clouds around the equator in areas where very thin high cirrus is common. The MAE-difference (Figure 6) shows that the NN-AVHRR is better than MODIS-C6 in most parts of the globe, with the greatest benefit observed closer to the poles. At a few isolated locations MODIS-C6 is better than NN-AVHRR.

## 4.7   Future work and challenges

Only near nadir satellite zenith angles were used for training. This might limit the performance for the neural networks at other satellite zenith angles. The NN-MetImage network using the $CO_2$ channel at 13.3 μm shows strong satellite zenith angle dependence and is not useful for higher satellite zenith angles. A solution to train networks to perform better at higher satellite zenith angles could be to include MODIS data from satellite Terra co-located with CALIPSO in the training data, as they will get matches at any satellite zenith angle although only at high latitudes. As latitude is not used as a variable, data for higher satellite zenith angles included for high latitude regions could help also in other regions. However it might be that the high latitude matches will not help the network if the variety of weather situations and cloud top heights at high latitudes is too small. Radiative transfer calculations for the $CO_2$-channels for different satellite-zenith angles could be another way to improve the performance for higher satellite-zenith angles.

Several technical parameters influence the performance of the neural network, for example: learning rate, learning rate decay, momentum, number of layers, number of neurons, weight initialization function and early stopping criteria. For several combinations tested, the differences were in the order of a few hPa. Networks tested using two hidden layers were found to perform better than those using only one hidden layer. We did train one network with less neurons and one with more layers and neurons with the same variables as NN-AVHRR. The network with fewer neurons in the two hidden layers (20/15) was 1 hPa worse. The network with more neurons in three layers (30/45/45) was 2.5hPa better than NN-AVHRR but also took 5 times as long time to retrieve pressure. The best technical parameters and network setup to use could be further investigated.

The NN-CTTH algorithm currently has no pixel specific error estimate. The MAE provides a constant error estimate (the same for all pixels). However for some clouds the height retrieval is more difficult, e.g. thin clouds and sub-pixel clouds. Further work to include pixel specific error estimates could be valuable.

Neural networks can behave unexpectedly for unseen data. By using a large training dataset and early stopping the risk for unexpected behaviour is decreased. Also the risk for unexpected results in a neural network algorithm can be a fair price to pay given the significant improvements when compared to the current algorithms. The training of neural networks requires reference data (truth). For optimal performance a neural network approach for upcoming new sensors (e.g. MERSI-2, MetImage) being launched when data from CALIPSO or CloudSat are no longer available, either another truth is needed or a method to robustly transform network trained for one sensor to other sensors is needed. A way forward could be to include variables with

radiative transfer calculations of cloud free brightness temperatures and brightness temperature differences. Further work is needed to test how the networks trained for the MODIS sensor perform for AVHRR, AVHRR1, VIIRS and other sensors. Our results show that networks can be trained using only the channels available on AVHRR, but they might need to be retrained with actual AVHRR data as the spectral response functions of the channels differ. The spectral response functions also dif-
fer between different AVHRR instruments, and more investigations are needed to see how networks trained for one AVHRR instrument will perform for other AVHRR instruments.

The results here are valid for the MODIS imager on the polar orbiting satellite Aqua. However nothing in the method restricts it to polar orbiting satellites. The method should be applicable for imagers like SEVIRI, which has the two most important channels at 11 μm and 12 μm, on geostationary satellites. However the network trained on MODIS data might need to be
retrained with SEVIRI data to get the best performance as the spectral response functions between SEVIRI and MODIS differ.

## 5   Conclusions

The neural network approach shows high potential to improve cloud height retrievals. The NN-CTTH (for all trained neural networks) is better in terms of MAE in meters than both PPS-v2014 and the MODIS Collection 6. This is seen for validation with CALIOP and CPR (CloudSat) and for low, medium, high level clouds. The neural networks also show best MAE for all
cloud types except *altostratus (opaque)* for which PPS-v2014 is better than some of the neural networks. The neural networks show an overall improvement of mean absolute error (MAE) from 400 m and up to 1km. Considering overall performance in terms of IQR, RMSE, SD, $PE_{0.25}$, $PE_{0.5}$, $PE_1$, $PE_2$, median, mode and bias the neural network performs better than both the reference algorithms both when validated with CALIOP and CPR (CloudSat). In the validation with CALIOP the neural networks have between 7 and 20 percentages more retrievals with absolute errors smaller than 250 m compared to the reference
algorithms. Considering low, medium and high levels separately the neural networks perform better or for some cases in line with the best of the two reference algorithms in terms of MAE, IQR, $PE_{0.25}$, $PE_{0.5}$, $PE_1$, median and mode. This indicates that the neural networks have well centered, narrow error distributions with large amount of retrievals with small errors.

The two reference algorithms have been shown to have different strenghts MODIS-C6 validated with CPR (CloudSat) for high clouds shows a well centered and narrow error distribution in line with (and better than some of) the neural networks,
although the MAE is higher for MODIS-C6. PPS-v2014 validated with CALIOP for the cloud type *altostratus (opaque)* show scores in line with (and better than some of) the neural networks.

The error distributions for the cloud top height retrievals were found to be skewed for all algorithms considered in the paper, especially for low level clouds. It was examplified why the bias and SD should be interpreted with caution and how they can easily be misinterpreted. The median and mode where found to be better measures of tendency than the bias. The IQR and
MAE were found to better describe the spread of the errors, compared to SD and RMSE, as the absolute values of the largest errors are not the most interesting. Measuring the amount of absolute error above for example 1km ($PE_1$) was found to provide valuable information on the amount of large/small errors and useful retrievals.

The neural network algorithms are also useful for instruments with fewer channels than MODIS, including the channels available for AVHRR1. This is important for climate data records which include AVHRR1 data to produce a long, continuous time series. Including variables with information on neighbouring pixel values was very important to get good results about and 40 % of the improvement of MAE for the cloud top pressure retrieval for NN-AVHRR was due to the variables with neighbouring pixels. The networks trained using only two IR-channels at 11 µm and 12 µm or 3.7 µm showed the most robust performance at higher satellite zenith angles. Including more IR channels does improve results for nadir observations, but degrades performance at higher satellite zenith angles.

A neural network cloud top pressure, temperature and height algorithm will be be part of the PPS-v2018 release. The PPS software package is accessible via the NWC SAF site nwc-saf.eumetsat.int.

*Author contributions.* All authors contributed to designing the study. Claudia Adok and Nina Håkansson wrote the code and carried out the experiments. Nina Håkansson drafted the manuscript and prepared the figures and tables. All authors discussed results and revised the manuscript.

*Competing interests.* The authors declare that they have no conflict of interest.

*Acknowledgements.* The authors acknowledge that the work was mainly funded by EUMETSAT. The CALIOP data were obtained from the NASA Langley Research Center Atmospheric Science Data Center. The CloudSat Data Processing Center (DPC) and Science Teams are further acknowledged for providing CloudSat datasets. NWP data were downloaded from ECMWF. The MODIS/Aqua dataset was acquired from the Level-1 & Atmosphere Archive and Distribution System (LAADS) Distributed Active Archive Center (DAAC), located in the Goddard Space Flight Center in Greenbelt, Maryland (https://ladsweb.nascom.nasa.gov/). The authors thank Thomas Heinemann (EUMETSAT) for suggesting adding CPR (CloudSat) as an independent validation truth.

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

**Table 1.** MODIS data from 2010 used for training and validation of the neural networks.

| Dataset | Days used |
| --- | --- |
| Training | $1^{st}$ January March July September |
| | $14^{th}$ February April May |
| | $14^{th}$ August October December |
| Validation during training | $1^{st}$ May |
| | $14^{th}$ March July November |
| Testing under development | $1^{st}$ November |
| | $14^{th}$ January June September |
| Final validation | $1^{st}$ February April June |
| | $1^{st}$ August October December |

**Table 2.** Description of variable types used to train the neural networks.

| Variable type | Variable names | Note |
| --- | --- | --- |
| Surface pressure | $P_S$ | Max pressure for pixel |
| NWP temperatures at surface, 950, 850, 700, 500, 250 hPa | $T_S, T_{950}, T_{850}, T_{700}, T_{500}, T_{250}$ | BT to pressure conversion |
| NWP column integrated water vapour | $Ciwv$ | Expected BT differences |
| Brightness Temperature (BT) for 11 µm or 12 µm | $B_{11}, B_{12}$ | Opaque temperature |
| BT for water vapour channels at 6.7 µm or 7.3 µm | $B_{6.7}, B_{7.3}$ | High or low |
| BT for CO$_2$ channel at 13.3 µm | $B_{13.3}$ | High or low |
| BT differences | $B_{11} - B_{12}, B_{11} - B_{3.7}, B_{8.5} - B_{11}$ | Opacity or phase |
| BT differences *to* warmest/coldest neighbour | $B_{12}^W - B_{12}, B_{12}^C - B_{12}$ or | Edge or thin |
| | $B_{11}^W - B_{11}, B_{11}^C - B_{11}$ | Edge or thin |
| BT differences *for* warmest/coldest neighbour | $B_{11}^W - B_{12}^W, B_{11}^C - B_{12}^C$ or | Opacity |
| | $B_{11}^W - B_{3.7}^W, B_{11}^C - B_{3.7}^C$ | Opacity |
| Texture: standard deviation of variable for 5x5 pixels | $S_{B_{11}-B_{12}}, S_{B_{11}}, S_{B_{3.7}}$ | Edge or thin |

**Table 3.** Description of the different networks. See Table 2 for explanation of the variables. The NWP variables: $P_S$, $T_S$, $T_{950}$, $T_{850}$, $T_{700}$, $T_{500}$, $T_{250}$ are used in all networks.

| Network name | Network specific variables |
|---|---|
| NN-NWP | $Ciwv$ |
| NN-OPAQUE | $B_{12}$ |
| NN-BASIC | $B_{12}, B_{11} - B_{12},$ |
| NN-BASIC-CIWV | $B_{12}, B_{11} - B_{12}, Ciwv,$ |
| NN-AVHRR | $B_{12}, B_{11} - B_{12}, Ciwv,$ |
| | $B_{11}^W - B_{12}^W, B_{11}^C - B_{12}^C,$ |
| | $B_{12}^W - B_{12}, B_{12}^C - B_{12},$ |
| | $S_{B_{11}-B_{12}}, S_{B_{11}},$ |
| NN-VIIRS | $B_{12}, B_{11} - B_{12}, Ciwv,$ |
| | $B_{11}^W - B_{12}^W, B_{11}^C - B_{12}^C,$ |
| | $B_{12}^W - B_{12}, B_{12}^C - B_{12},$ |
| | $S_{B_{11}-B_{12}}, S_{B_{11}},$ |
| | $B_{8.5} - B_{11}$ |
| NN-MERSI-2 | $B_{12}, B_{11} - B_{12}, Ciwv,$ |
| | $B_{11}^W - B_{12}^W, B_{11}^C - B_{12}^C,$ |
| | $B_{12}^W - B_{12}, B_{12}^C - B_{12},$ |
| | $S_{B_{11}-B_{12}}, S_{B_{11}},$ |
| | $B_{8.5} - B_{11}, B_{7.3}$ |
| NN-MetImage-NoCO$_2$ | $B_{12}, B_{11} - B_{12}, Ciwv,$ |
| | $B_{11}^W - B_{12}^W, B_{11}^C - B_{12}^C,$ |
| | $B_{12}^W - B_{12}, B_{12}^C - B_{12},$ |
| | $S_{B_{11}-B_{12}}, S_{B_{11}},$ |
| | $B_{8.5} - B_{11}, B_{7.3}, B_{6.7}$ |
| NN-MetImage | $B_{12}, B_{11} - B_{12}, Ciwv,$ |
| | $B_{11}^W - B_{12}^W, B_{11}^C - B_{12}^C,$ |
| | $B_{12}^W - B_{12}, B_{12}^C - B_{12},$ |
| | $S_{B_{11}-B_{12}}, S_{B_{11}},$ |
| | $B_{8.5} - B_{11}, B_{7.3}, B_{6.7}, B_{13.3}$ |
| NN-AVHRR1 | $B_{11}, B_{11} - B_{3.7}, Ciwv,$ |
| | $B_{11}^W - B_{3.7}^W, B_{11}^C - B_{3.7}^C$ |
| | $B_{11}^W - B_{11}, B_{11}^C - B_{11}$ |
| | $S_{B_{3.7}}, S_{B_{11}}$ |

**Table 4.** Description of the imager channels used for the different algorithms. For MODIS-C6 channels used indirectly, to determine if $CO_2$-slicing should be applied, are noted with brackets.

| Imager channel: Network name | $B_{3.7}$ | $B_{6.7}$ | $B_{7.3}$ | $B_{8.5}$ | $B_{11}$ | $B_{12}$ | $B_{13.3}$ | $B_{13.6}$ | $B_{13.9}$ | $B_{14.2}$ |
|---|---|---|---|---|---|---|---|---|---|---|
| PPS-v2014 | | | | | x | x | | | | |
| MODIS-C6 | | | (x) | (x) | x | (x) | x | x | x | x |
| NN-NWP | | | | | | | | | | |
| NN-OPAQUE | | | | | | x | | | | |
| NN-BASIC | | | | | x | x | | | | |
| NN-BASIC-CIWV | | | | | x | x | | | | |
| NN-AVHRR | | | | | x | x | | | | |
| NN-VIIRS | | | | x | x | x | | | | |
| NN-MERSI-2 | | | x | x | x | x | | | | |
| NN-MetImage-NoCO$_2$ | | x | x | x | x | x | | | | |
| NN-MetImage | | x | x | x | x | x | x | | | |
| NN-AVHRR1 | x | | | | x | | | | | |

**Table 5.** Mean absolute error (MAE) for different algorithms compared to CALIOP top layer pressure. The final validation dataset (see Table 1), containing 1832432 pixels (45 % high, 39 % low and 16 % medium level clouds) is used. Pixels with valid pressure for PPS-v2014, MODIS-C6, and CALIOP are considered. The low, medium and high classes are from CALIOP feature classification flag.

| | MAE [hPa] | | | |
|---|---|---|---|---|
| | all | low | medium | high |
| PPS-v2014 | 122.6 | 80.2 | 88.0 | 172.9 |
| MODIS-C6 | 123.9 | 90.7 | 139.8 | 147.3 |
| NN-NWP | 191.7 | 141.7 | 110.3 | 265.8 |
| NN-OPAQUE | 113.2 | 82.1 | 105.0 | 143.8 |
| NN-BASIC | 93.9 | 67.7 | 92.8 | 117.6 |
| NN-BASIC-CIWV | 92.1 | 67.5 | 91.3 | 114.2 |
| NN-AVHRR | 72.4 | 55.4 | 67.6 | 89.2 |
| NN-VIIRS | 65.9 | 50.5 | 59.3 | 81.9 |
| NN-MERSI2 | 61.4 | 48.2 | 52.1 | 76.6 |
| NN-MetImage-NoCO$_2$ | 60.3 | 47.1 | 54.5 | 74.1 |
| NN-MetImage | 54.2 | 44.5 | 51.6 | 63.8 |
| NN-AVHRR1 | 76.1 | 54.7 | 69.9 | 97.3 |

**Table 6.** Statistic measures for the error distributions for all clouds. For all measures except skewness it is the case that values closer to zero are better. The statistics are calculated for 1198599 matches for CPR (CloudSat) and 1803335 matches for CALIOP. A small amount 0.2% of the matches were excluded because of missing height or pressure below 70 hPa for any of the algorithms. $PE_X$ describes percentage of absolute errors above X km, see Equation 1.

| | MAE | IQR | RMSE | SD[1] | $PE_{0.25}$ | $PE_{0.5}$ | $PE_1$ | $PE_2$ | median | mode | bias[1] | skew |
| --- | --- | --- | --- | --- | --- | --- | --- | --- | --- | --- | --- | --- |
| | [m] | [m] | [m] | [m] | [%] | [%] | [%] | [%] | [m] | [m] | [m] | |
| CALIOP all clouds | | | | | | | | | | | | |
| PPS-v2014 | 2095 | 2832 | 3188 | 2832 | 82 | 69 | 54 | 29 | -639 | -118 | -1465 | -1.0 |
| MODIS-C6 | 1923 | 2177 | 3105 | 2883 | 85 | 72 | 51 | 23 | -612 | -262 | -1153 | -1.5 |
| NN-AVHRR | 1300 | 1326 | 2234 | 2197 | 73 | 56 | 36 | 14 | 50 | 106 | -405 | -1.8 |
| NN-VIIRS | 1187 | 1189 | 2114 | 2074 | 71 | 52 | 33 | 12 | 28 | 100 | -410 | -1.9 |
| NN-MERSI-2 | 1120 | 1107 | 2039 | 1996 | 69 | 50 | 30 | 11 | -2 | 73 | -420 | -2.0 |
| NN-MetImage-NoCO$_2$ | 1091 | 1040 | 2009 | 1966 | 68 | 48 | 29 | 11 | -49 | 44 | -416 | -2.0 |
| NN-MetImage | 979 | 909 | 1840 | 1817 | 65 | 46 | 26 | 9 | -17 | 15 | -294 | -1.9 |
| NN-AVHRR1 | 1383 | 1547 | 2354 | 2281 | 75 | 58 | 38 | 16 | -42 | 50 | -584 | -1.8 |
| CPR (CloudSat) all clouds | | | | | | | | | | | | |
| PPS-v2014 | 1744 | 2255 | 2432 | 2160 | 87 | 74 | 56 | 24 | -833 | -426 | -1118 | -0.1 |
| MODIS-C6 | 1692 | 1928 | 2607 | 2533 | 84 | 70 | 48 | 20 | -375 | -259 | -614 | -0.1 |
| NN-AVHRR | 1262 | 1473 | 1928 | 1923 | 77 | 61 | 41 | 14 | 88 | -141 | 143 | 0.2 |
| NN-VIIRS | 1207 | 1368 | 1901 | 1896 | 76 | 58 | 38 | 13 | 69 | -146 | 137 | 0.5 |
| NN-MERSI-2 | 1120 | 1275 | 1793 | 1788 | 75 | 56 | 35 | 11 | 40 | -201 | 136 | 0.5 |
| NN-MetImage-NoCO$_2$ | 1146 | 1315 | 1834 | 1828 | 76 | 57 | 35 | 12 | 9 | -218 | 147 | 0.7 |
| NN-MetImage | 1170 | 1421 | 1865 | 1843 | 76 | 58 | 37 | 11 | 84 | -243 | 285 | 0.9 |
| NN-AVHRR1 | 1281 | 1523 | 1953 | 1953 | 79 | 63 | 41 | 14 | 30 | -128 | -14 | 0.0 |

[1] Interpret bias and SD with caution as distributions are non-Gaussian. Bias is not located at the center of the distribution.

**Table 7.** Statistic measures for the error distributions for low level clouds. For all measures except skewness it is the case that values closer to zero are better. The statistics are calculated for 328015 matches for CPR (CloudSat) and 709434 matches for CALIOP. The low class comes from CALIOP feature classification flag (class 0, 1, 2 and 3) and for CPR (CloudSat) it is the pixels with heights lower or exactly at the NWP height at 680 hPa. $PE_X$ describes percentage of absolute errors above X km, see Equation 1.

| | MAE | IQR | RMSE | SD[1] | $PE_{0.25}$ | $PE_{0.5}$ | $PE_1$ | $PE_2$ | median | mode | bias[1] | skew |
|---|---|---|---|---|---|---|---|---|---|---|---|---|
| | [m] | [m] | [m] | [m] | [%] | [%] | [%] | [%] | [m] | [m] | [m] | |
| Low level clouds CALIOP | | | | | | | | | | | | |
| PPS-v2014 | 847 | 1035 | 1469 | 1436 | 68 | 47 | 27 | 5 | -46 | -117 | 312 | 3.0 |
| MODIS-C6 | 952 | 1230 | 1576 | 1561 | 78 | 58 | 29 | 6 | -17 | -150 | 219 | 2.9 |
| NN-AVHRR | 586 | 584 | 1121 | 1027 | 56 | 31 | 14 | 3 | 215 | 101 | 449 | 4.0 |
| NN-VIIRS | 533 | 515 | 1080 | 1006 | 52 | 27 | 11 | 3 | 182 | 126 | 391 | 4.8 |
| NN-MERSI-2 | 509 | 490 | 1063 | 998 | 49 | 25 | 10 | 3 | 159 | 86 | 365 | 4.8 |
| NN-MetImage-NoCO$_2$ | 499 | 504 | 1068 | 1024 | 48 | 24 | 10 | 3 | 98 | 40 | 303 | 4.9 |
| NN-MetImage | 476 | 450 | 1103 | 1069 | 45 | 21 | 8 | 3 | 74 | 14 | 271 | 5.4 |
| NN-AVHRR1 | 574 | 646 | 1045 | 969 | 58 | 33 | 13 | 3 | 197 | 49 | 391 | 3.8 |
| Low level clouds CPR (CloudSat) | | | | | | | | | | | | |
| PPS-v2014 | 949 | 1197 | 1571 | 1556 | 78 | 56 | 29 | 5 | -173 | -413 | 211 | 2.8 |
| MODIS-C6 | 1192 | 1335 | 2145 | 2097 | 79 | 60 | 33 | 9 | 46 | -110 | 450 | 2.9 |
| NN-AVHRR | 743 | 685 | 1595 | 1532 | 56 | 31 | 16 | 6 | 16 | -132 | 443 | 3.8 |
| NN-VIIRS | 739 | 637 | 1690 | 1633 | 55 | 30 | 15 | 6 | -6 | -139 | 432 | 4.2 |
| NN-MERSI-2 | 721 | 605 | 1652 | 1602 | 55 | 28 | 14 | 6 | -31 | -181 | 403 | 4.1 |
| NN-MetImage-NoCO$_2$ | 742 | 608 | 1670 | 1637 | 60 | 31 | 13 | 6 | -105 | -255 | 328 | 4.2 |
| NN-MetImage | 773 | 578 | 1813 | 1775 | 58 | 30 | 13 | 6 | -102 | -217 | 369 | 4.1 |
| NN-AVHRR1 | 827 | 852 | 1676 | 1602 | 64 | 38 | 18 | 7 | 48 | -198 | 491 | 3.6 |

[1] Interpret bias and SD with caution as distributions are non-Gaussian. Bias is not located at the center of the distribution.

**Table 8.** Statistic measures for the error distributions for medium level clouds. For all measures except skewness it is the case that values closer to zero are better. The statistics are calculted for 244885 matches for CPR (CloudSat) and 295186 matches for CALIOP. The high class comes from CALIOP feature classification flag (class 4 and 5) and for CPR (CloudSat) it is the pixels with heights between the NWP height at 440 hPa and 680 hPa. $PE_X$ describes percentage of absolute errors above X km, see Equation 1.

| | MAE | IQR | RMSE | SD[1] | $PE_{0.25}$ | $PE_{0.5}$ | $PE_1$ | $PE_2$ | median | mode | bias[1] | skew |
|---|---|---|---|---|---|---|---|---|---|---|---|---|
| | [m] | [m] | [m] | [m] | [%] | [%] | [%] | [%] | [m] | [m] | [m] | |
| *Medium level clouds CALIOP* | | | | | | | | | | | | |
| PPS-v2014 | 1121 | 1600 | 1651 | 1614 | 78 | 59 | 37 | 12 | -68 | 124 | -348 | 0.2 |
| MODIS-C6 | 1759 | 2590 | 2304 | 2192 | 87 | 76 | 60 | 27 | -654 | 205 | -708 | 0.6 |
| NN-AVHRR | 969 | 1243 | 1394 | 1339 | 78 | 59 | 34 | 7 | 304 | 273 | 387 | 0.8 |
| NN-VIIRS | 832 | 1048 | 1227 | 1206 | 74 | 53 | 28 | 5 | 186 | 23 | 223 | 0.7 |
| NN-MERSI-2 | 731 | 935 | 1102 | 1093 | 70 | 47 | 23 | 4 | 83 | 16 | 144 | 0.9 |
| NN-MetImage-NoCO$_2$ | 762 | 984 | 1148 | 1145 | 71 | 49 | 24 | 4 | 28 | -1 | 86 | 1.1 |
| NN-MetImage | 714 | 905 | 1091 | 1090 | 69 | 46 | 22 | 3 | 4 | -63 | 36 | 1.1 |
| NN-AVHRR1 | 980 | 1330 | 1381 | 1364 | 79 | 61 | 35 | 7 | 187 | 176 | 213 | 0.5 |
| *Medium level clouds CPR (CloudSat)* | | | | | | | | | | | | |
| PPS-v2014 | 1364 | 1978 | 1927 | 1858 | 82 | 66 | 46 | 18 | -300 | 53 | -512 | 0.5 |
| MODIS-C6 | 1909 | 2698 | 2532 | 2475 | 88 | 78 | 62 | 30 | -597 | 69 | -534 | 0.9 |
| NN-AVHRR | 1215 | 1541 | 1817 | 1770 | 81 | 64 | 40 | 12 | 209 | -113 | 409 | 1.2 |
| NN-VIIRS | 1139 | 1325 | 1788 | 1760 | 77 | 59 | 36 | 11 | 114 | -81 | 310 | 1.5 |
| NN-MERSI-2 | 1059 | 1203 | 1706 | 1686 | 75 | 55 | 32 | 10 | 15 | -150 | 264 | 1.7 |
| NN-MetImage-NoCO$_2$ | 1091 | 1259 | 1752 | 1740 | 76 | 57 | 33 | 10 | -44 | -154 | 205 | 1.8 |
| NN-MetImage | 1113 | 1217 | 1832 | 1818 | 75 | 56 | 33 | 11 | -45 | -174 | 225 | 1.9 |
| NN-AVHRR1 | 1221 | 1591 | 1776 | 1751 | 81 | 65 | 41 | 13 | 146 | -25 | 301 | 1.0 |

[1] Interpret bias and SD with caution as distributions are non-Gaussian. Bias is not located at the center of the distribution.

**Table 9.** Statistic measures for the error distributions for high level clouds. For all measures except skewness it is the case that values closer to zero are better. The statistics are calculated for 625699 matches for CPR (CloudSat) and 798715 matches for CALIOP. The high class comes from CALIOP feature classification flag (class 6 and 7) and for CPR (CloudSat) it is the pixels with heights higher or exactly at the NWP height at 440 hPa. $PE_X$ describes percentage of absolute errors above X km, see Equation 1.

| | MAE | IQR | RMSE | SD[1] | $PE_{0.25}$ | $PE_{0.5}$ | $PE_1$ | $PE_2$ | median | mode | bias[1] | skew |
|---|---|---|---|---|---|---|---|---|---|---|---|---|
| | [m] | [m] | [m] | [m] | [%] | [%] | [%] | [%] | [m] | [m] | [m] | |
| *High level clouds CALIOP* | | | | | | | | | | | | |
| PPS-v2014 | 3564 | 3367 | 4475 | 2842 | 96 | 92 | 84 | 57 | -2918 | -1897 | -3456 | -0.9 |
| MODIS-C6 | 2846 | 3095 | 4196 | 3342 | 92 | 84 | 68 | 36 | -1586 | -917 | -2537 | -1.5 |
| NN-AVHRR | 2057 | 2775 | 3072 | 2704 | 87 | 76 | 57 | 27 | -799 | -130 | -1457 | -1.4 |
| NN-VIIRS | 1899 | 2459 | 2916 | 2581 | 86 | 74 | 53 | 23 | -716 | -18 | -1356 | -1.6 |
| NN-MERSI-2 | 1807 | 2258 | 2818 | 2486 | 85 | 72 | 51 | 21 | -705 | -192 | -1326 | -1.7 |
| NN-MetImage-NoCO$_2$ | 1739 | 2134 | 2760 | 2464 | 84 | 70 | 48 | 20 | -606 | -248 | -1242 | -1.8 |
| NN-MetImage | 1524 | 1906 | 2476 | 2298 | 83 | 67 | 44 | 16 | -360 | -83 | -920 | -2.0 |
| NN-AVHRR1 | 2250 | 2913 | 3292 | 2791 | 89 | 79 | 61 | 30 | -1099 | -475 | -1746 | -1.3 |
| *High level clouds CPR (CloudSat)* | | | | | | | | | | | | |
| PPS-v2014 | 2309 | 2384 | 2930 | 2092 | 93 | 87 | 74 | 36 | -1789 | -1428 | -2052 | -0.5 |
| MODIS-C6 | 1869 | 2142 | 2845 | 2578 | 86 | 73 | 51 | 22 | -614 | -506 | -1203 | -1.2 |
| NN-AVHRR | 1553 | 2244 | 2121 | 2118 | 87 | 75 | 54 | 19 | 143 | 348 | -117 | -0.6 |
| NN-VIIRS | 1479 | 2095 | 2043 | 2041 | 86 | 73 | 52 | 17 | 168 | 332 | -85 | -0.7 |
| NN-MERSI-2 | 1353 | 1876 | 1894 | 1893 | 85 | 71 | 48 | 14 | 177 | 326 | -54 | -0.9 |
| NN-MetImage-NoCO$_2$ | 1379 | 1843 | 1944 | 1944 | 85 | 71 | 48 | 15 | 219 | 292 | 29 | -0.7 |
| NN-MetImage | 1399 | 1871 | 1904 | 1885 | 87 | 74 | 52 | 14 | 463 | 511 | 265 | -0.8 |
| NN-AVHRR1 | 1542 | 2275 | 2145 | 2107 | 87 | 74 | 53 | 19 | -67 | 281 | -403 | -0.8 |

[1] Interpret bias and SD with caution as distributions are non-Gaussian. Bias is not located at the center of the distribution.

**Table 10.** Mean absolute error (MAE) and median in meters for different algorithms compared to CALIOP top layer altitude. The final validation dataset (see Table 1), containing 1803335 pixels (5 % low overcast (transparent), 12 % low overcast opaque, 19 % transition stratocumulus, 2 % low, broken cumulus, 7 % altocumulus (transparent), 8 % altostratus (opaque), 30 % cirrus (transparent) and 14 % deep convective (opaque)), where all algorithms had a cloud top height is used. The cloud types are from CALIOP feature classification. $PE_{0.5}$ describes percentage of absolute errors above 0.5 km.

| | low overcast, transparent | low overcast, opaque | transition stratocumulus | low, broken cumulus | altocumulus (transparent) | altostratus (opaque) | cirrus (transparent) | deep convective (opaque) |
|---|---|---|---|---|---|---|---|---|
| **MAE [m]** | | | | | | | | |
| PPS-v2014 | 709 | 637 | 886 | 1695 | 1609 | 699 | 4343 | 1901 |
| MODIS-C6 | 903 | 1028 | 901 | 1058 | 2343 | 1254 | 3567 | 1308 |
| NN-AVHRR | 519 | 442 | 627 | 1027 | 1134 | 825 | 2608 | 883 |
| NN-VIIRS | 454 | 407 | 571 | 938 | 1011 | 678 | 2398 | 833 |
| NN-MERSI-2 | 408 | 381 | 550 | 946 | 900 | 584 | 2283 | 791 |
| NN-MetImage-NoCO$_2$ | 395 | 372 | 541 | 929 | 929 | 617 | 2210 | 734 |
| NN-MetImage | 365 | 364 | 509 | 912 | 885 | 565 | 1905 | 711 |
| NN-AVHRR1 | 516 | 448 | 617 | 911 | 1156 | 827 | 2847 | 977 |
| **median [m]** | | | | | | | | |
| PPS-v2014 | -183 | 50 | -90 | 220 | -633 | 63 | -3835 | -1716 |
| MODIS-C6 | -91 | 331 | -138 | -477 | -1953 | 85 | -2243 | -912 |
| NN-AVHRR | 223 | 160 | 241 | 380 | 109 | 410 | -1605 | 71 |
| NN-VIIRS | 185 | 143 | 201 | 315 | 7 | 279 | -1360 | 46 |
| NN-MERSI-2 | 160 | 116 | 177 | 313 | -34 | 145 | -1268 | -35 |
| NN-MetImage-NoCO$_2$ | 110 | 70 | 102 | 226 | -138 | 119 | -1133 | -19 |
| NN-MetImage | 53 | 46 | 86 | 214 | -163 | 87 | -787 | 125 |
| NN-AVHRR1 | 188 | 140 | 232 | 313 | -180 | 380 | -1895 | -82 |
| **PE$_{0.5}$ [%]** | | | | | | | | |
| PPS-v2014 | 46 | 38 | 49 | 67 | 76 | 44 | 95 | 87 |
| MODIS-C6 | 58 | 59 | 56 | 63 | 89 | 64 | 87 | 76 |
| NN-AVHRR | 33 | 23 | 34 | 47 | 67 | 53 | 83 | 60 |
| NN-VIIRS | 27 | 19 | 30 | 43 | 63 | 44 | 81 | 58 |
| NN-MERSI-2 | 24 | 17 | 28 | 42 | 58 | 37 | 79 | 57 |
| NN-MetImage-NoCO$_2$ | 22 | 16 | 27 | 40 | 59 | 40 | 78 | 53 |
| NN-MetImage | 20 | 15 | 23 | 37 | 58 | 36 | 74 | 52 |
| NN-AVHRR1 | 34 | 24 | 37 | 45 | 68 | 54 | 86 | 64 |

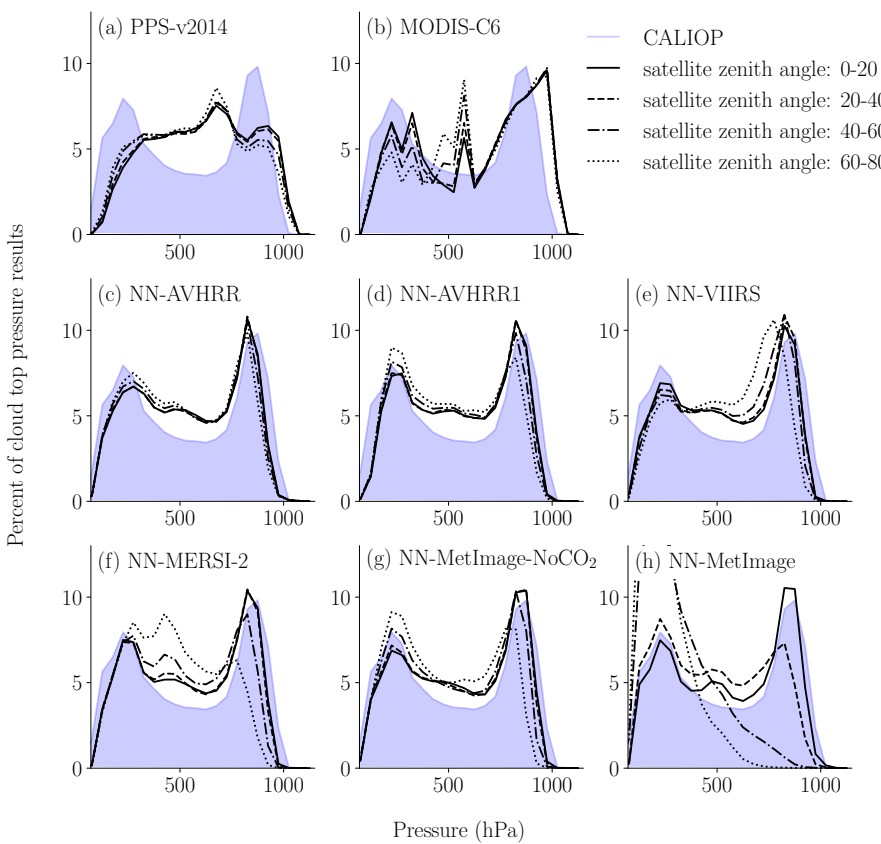

**Figure 1.** Retrieved pressure dependence on satellite zenih angle. CALIOP pressure distribution is shown in light blue. The percent of results are calculated in 50 hPa bins. The final validation dataset is used (see Table 1).

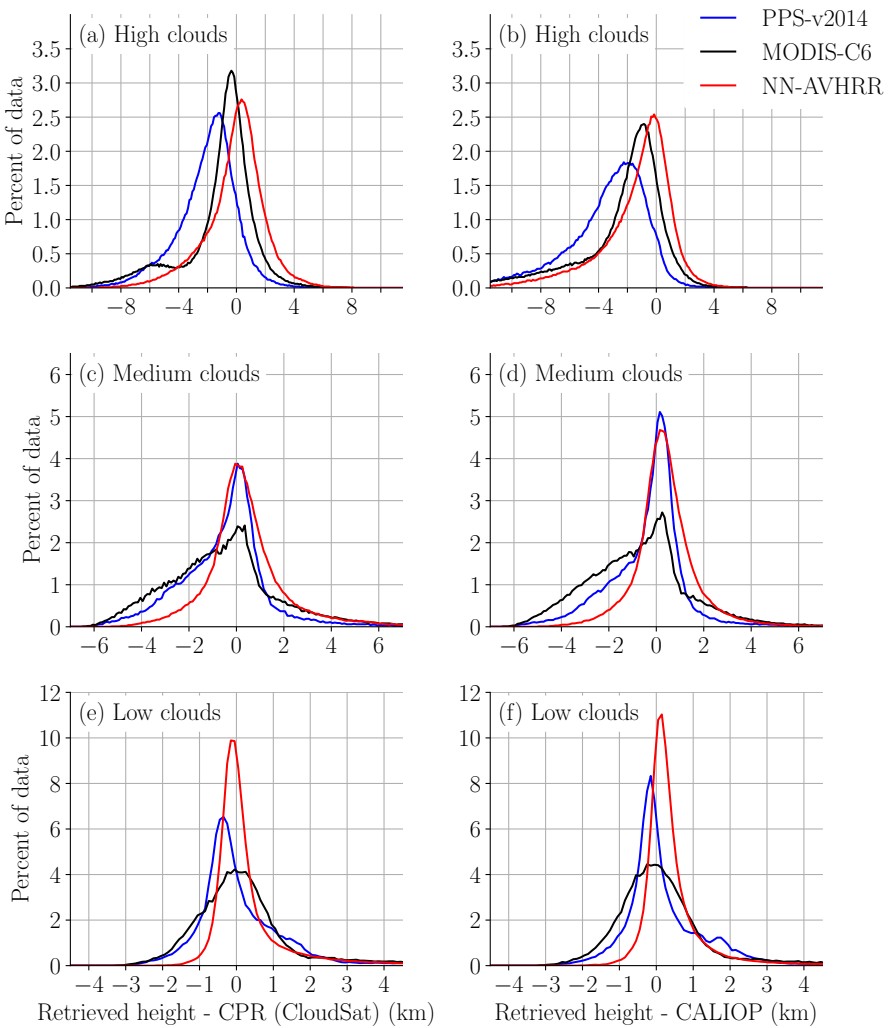

**Figure 2.** Error distribution compared to CPR (CloudSat) (left) and CALIOP (right). The percent of data is calculated in 0.1 km bins. For CALIOP the low, medium and high clouds are determined from CALIOP feature classification flag. For CPR (CloudSat) the low, medium, high clouds are determined from CPR (CloudSat) height compared to NWP geopotential height at 440 hPa and 680 hPa. The final validation dataset (see Table 1) where all algorithms had a height reported is used. Note that the values on the y-axis are dependent of the bin size. The peak at 11 % for NN-AVHRR in subplot (f), means that 11 % of the retrieved heights are between the CALIOP height and the CALIOP height + 0.1 km.

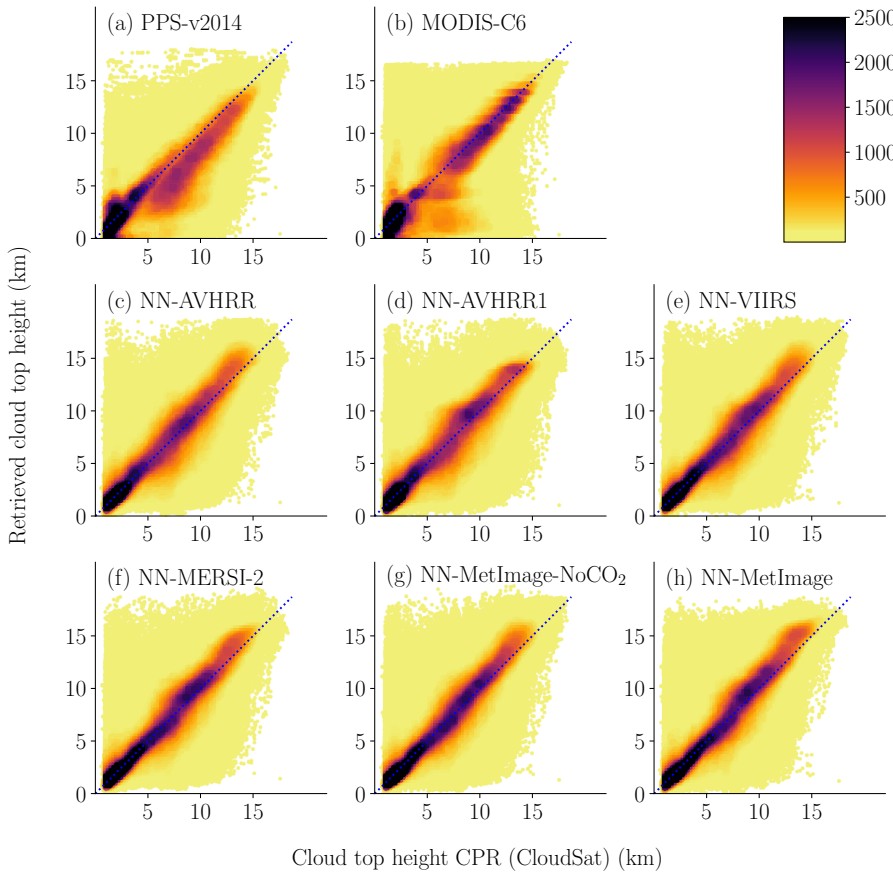

**Figure 3.** Scatters plot of the height for the neural networks and for the reference methods against CPR (CloudSat) height. The data were divided in bins of size 0.25 x 0.25 (km) for colour coding. The number of points in each bin determines the colour of the point. The final validation dataset (see Table 1) where all algorithms had a height reported is used. Two points where CPR (CloudSat) had a height above 22 km where excluded. A cloudy threshold of 30 % is used for CPR (CloudSat).

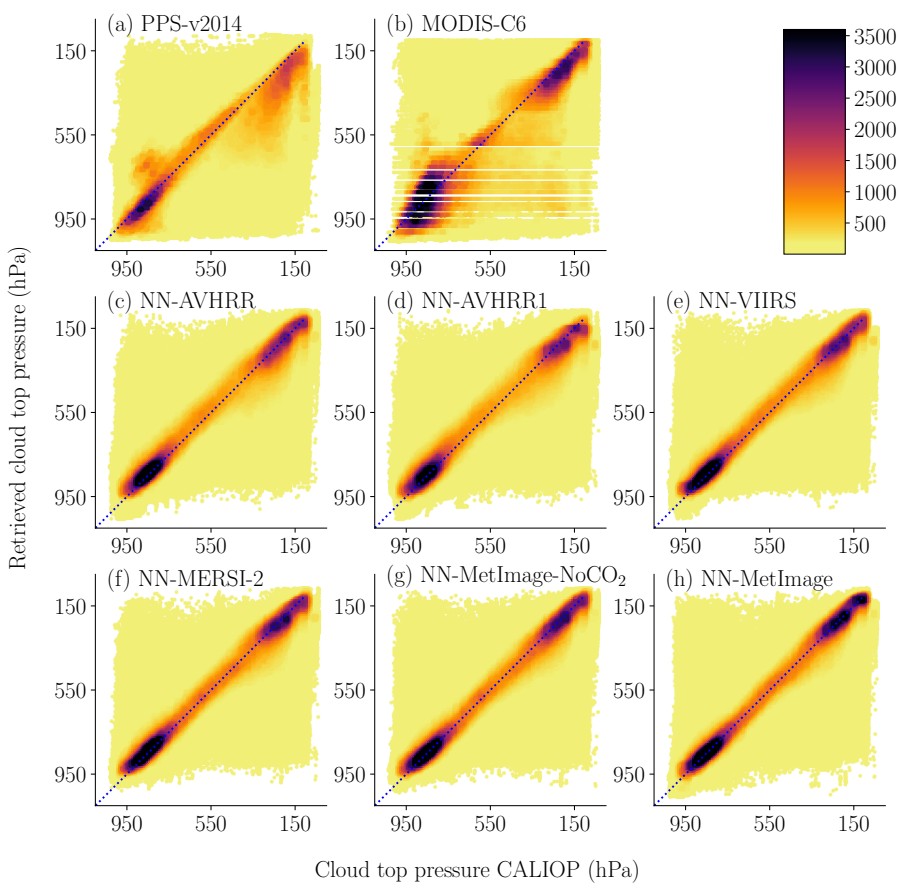

**Figure 4.** Scatter plots of the pressure for the neural networks and for the reference methods against CALIOP cloud top pressure. The data were divided in bins of size 10 x 10 (hPa) for colour coding. The number of points in each bin determines the colour of the point. The final validation dataset (see Table 1) where all algorithms had a height reported is used.

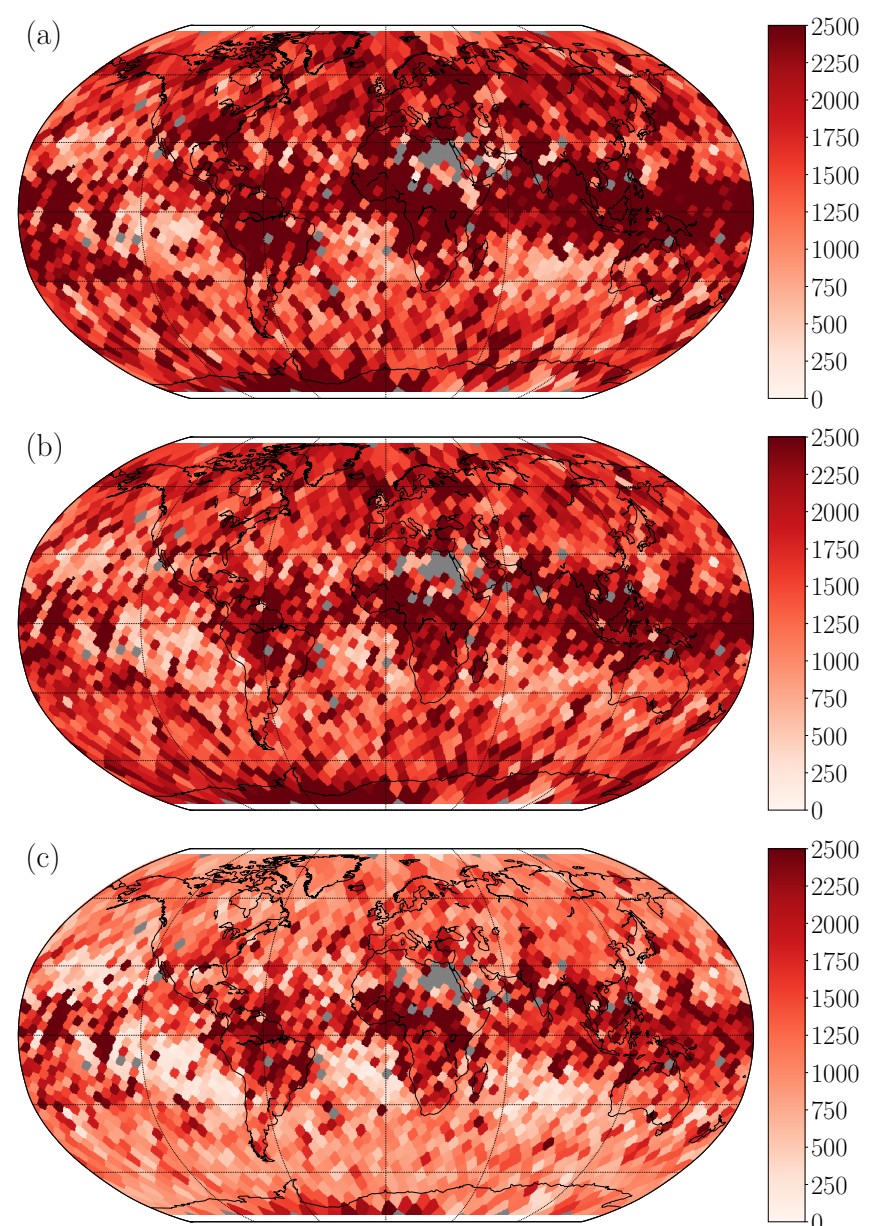

**Figure 5.** Mean absolute error in meters compared to CALIOP height. From the top a) PPS-v2014, b) MODIS-C6, and c) NN-AVHRR. Results are calculated for bins evenly spread out 250 km apart. Bins with less than 10 cloudy pixels are excluded (plotted in dark grey). The final validation and testing under development data (see Table 1) are included to get enough pixels.

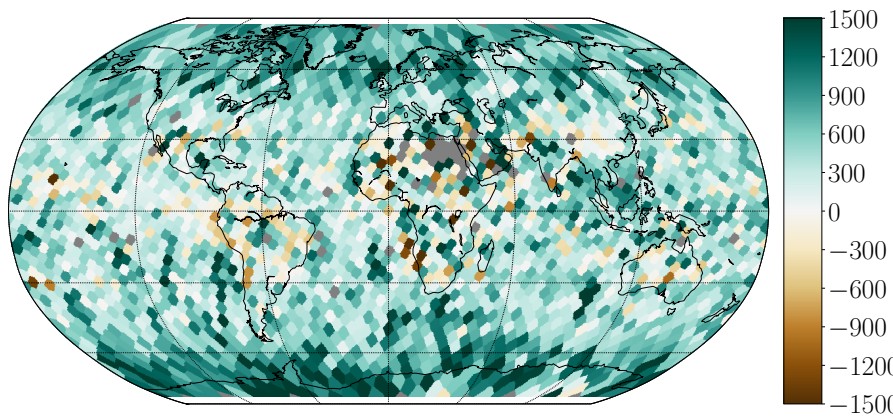

**Figure 6.** Mean absolute error difference in meters between MODIS-C6 and NN-AVHRR compared to CALIOP. Results are calculated for bins evenly spread out 250 km apart. Bins with less than 10 cloudy pixels are excluded (plotted in dark grey). Dark green means NN-AVHRR is 1.5 km better than MODIS-C6, dark brown means MODIS-C6 is 1.5 km better than NN-AVHRR. The final validation and testing under development data (see Table 1) are included to get enough pixels.

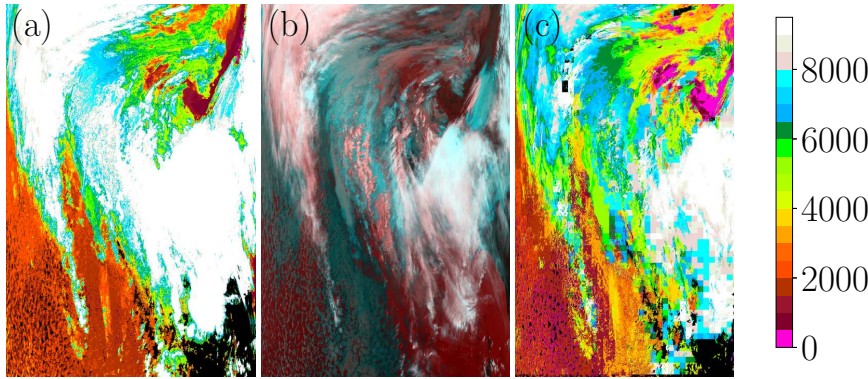

**Figure 7.** Comparing the cloud top height from the NN-AVHRR (left) to PPS-v2014 (right) with a RGB in the middle using channels at 3.7 μm, 11 μm, 12 μm. Notice that the NN-AVHRR is smoother, contain less nodata and that the small high ice clouds in the lower part of the figure are better captured. This is from MODIS on Aqua 14th of January 2010, 00:05UTC.

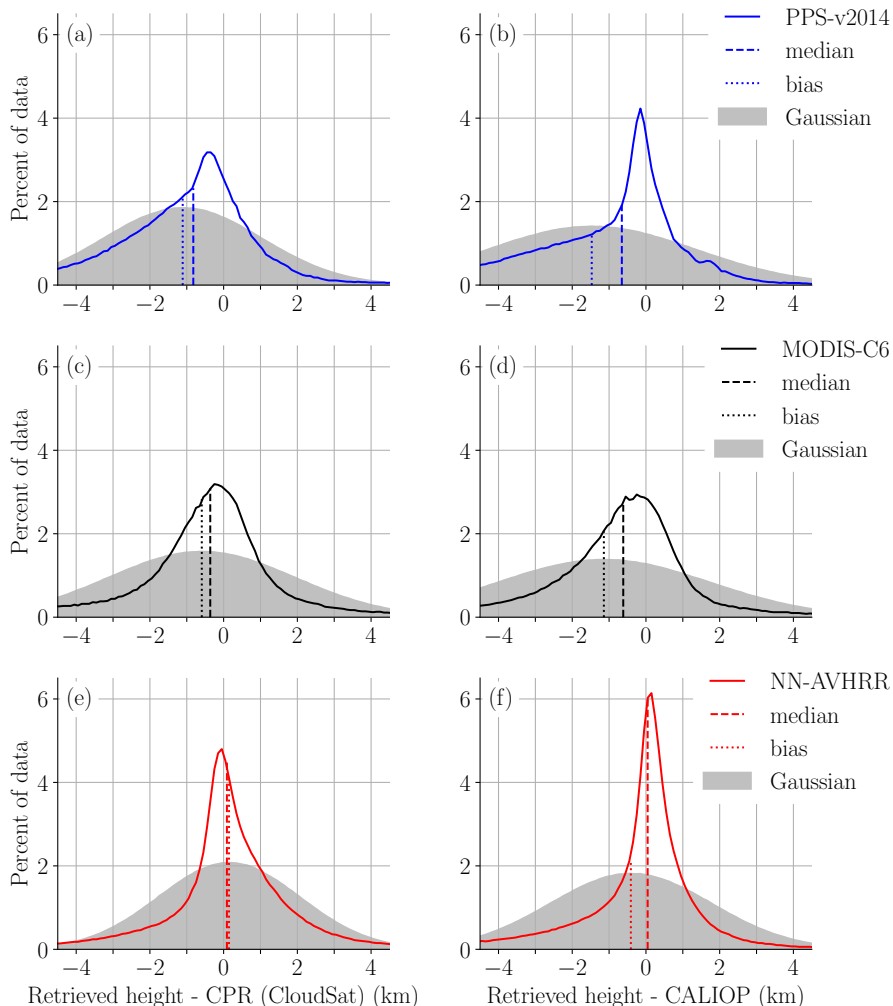

**Figure 8.** Error distribution compared to CPR (CloudSat) (left) and CALIOP (right) with biases and medians marked. The percent of data is calculated in 0.1 km bins. The final validation dataset (see Table 1) where all algorithms had a height reported is used. Note that the values on the y-axis are dependent of the bin size. The peak at 6 % for NN-AVHRR in subplot (f), means that 6 % of the retrieved heights are between the CALIOP height and the CALIOP height + 0.1 km. In grey the Gaussian distribution with the same bias and standard derivation is shown.