# Peer review of "Neural network cloud top pressure and height for MODIS"

_Atmospheric Measurement Techniques, 2017_

## Referee Comment (RC1) · Anonymous Referee #2 · 26 Feb 2018

Overall quality of the discussion paper ("general comments")

In the paper a novel retrieval of cloud top pressure and height using neural networks is presented. The presented retrieval technique is state of the art and an accurate technique. To account for different availabilities of channels on different satellites, a few modification of the neural network are investigated revealing the information content of the different channels. The new algorithms are compared to two reference algorithms, the CTTH algorithm of the NWCSAF PPS-v2014 and the MODIS collection 6 L2 height product. Additionally the algorithms is compared to CALIOP and CPR measurements. The quality of the algorithm is evaluated in terms of the mean absolute error (MAE). The improved quality of the results is impressive. From my point of view, I would request for at least another quality measure like standard deviation (similar to Tables 6 and 7). In

overall, good work!

Individual scientific questions/issues ("specific comments")

p1 line 23: CTH might also be used in data assimilation of atmospheric motion vectors. Introduction: A short description of the traditional technics to retrieve cloud top pressure and height could be added to the introduction. (or cite an overview paper like Hamann et al. "Remote sensing of cloud top pressure/height from SEVIRI: analysis of ten current retrieval algorithms." Atmospheric Measurement Techniques 7.9 (2014): 2839-2867.)

The introduction should motivate, why it is expected that using machine learning, in particular neural networks, could improve the expected results.

Merge chapter 2.1.1 into chapter 3.2. (and skip the sentence (p3 line 10) "The MODIS Collection 6 cloud product were used as an independent...", you said that before).

Chapter 2.2 Add a short sentence, why you chose the CALIOP 1km product and not also 5km or 10km product which are more sensitive to optically thin clouds.

Chapter 2.4 add the version number of the ECMWF model and add product name and version number of the OSISAF data used in this study.

You might consider to add the PPS-v2014 and MODIS C6 algorithm to table 3 and 4.

Please make the order of algorithms in table 3, 4 and 5 consistent.

p5 line 5: how often is a pressure lower than 70 hPa retrieved?

p5 line 10: Why did you choose this number of levels? Is it sufficient to use 6 levels to represent the boundary layer inversions or other small scale features?

p5 line 21: do you skip non cloudy pixels in the 5x5 pixel standard deviations?

p5 line 20: the $B\_3.7$ has a solar component. Did you correct for this during day/night?

p6 chapter 3.3.2: You chose to use specific days for training and others for validation.
Given that you only use a limited number of days, wouldn't it be more to randomly select independent pixels from all available dates for training, validation, and testing to represent a larger variety of weather situations?

p 6 line 19: Did you test other configurations that 30/15 neurons in the first/second layer? If yes, how was the performance?

Chapter 3.3.3: May batch size and momentum be changed during the training process?

p 7 line 27: consider to discuss the solar component of the 3.7 mue m channel. To my opinion this NN could perform better when corrected for that (e.g. adding the solar time as input variable).

p8 line 5: Maybe express it positively: All NN can reproduce a clear bi-modal pdf very similar to CALIPSO, the pdf of PPS-v2014 deviates from this shape . . .

p8 line 7: It is written "for the best performing network". Did you train several networks for one channel configuration? If so, could you describe the number of trained networks in chapter 3.2.2, please?

p 8, line 11: according to my table 6, the NN-MetImage is better than the NN-MetImage-NoCO2.

p 8, line 15: do you have an idea, why the MAE against CPR is larger than the MAE against CALIOP for NN-MetImage and NN-MetImage-NoCO2?

p 9, line 9: could you please describe a bit more in detail the differences seen in Figure 7?

Chapter 5 Discussion: Could you also comment on applying your NN technique on geostationary satellites? What would be the main differences/challenges?

A compact listing of purely technical corrections ("technical corrections": typing errors, etc.)

Please check consistent spelling of NWC SAF (e.g. p2 line 11) and NWCSAF (e.g. p1 line 19)

Please check consistent spelling of PPS-2018 (e.g. p1 line 19) and PPS-v2014 (e.g. p2 line 23)

Please check the space between numbers and units and the typeset of the units.

Try to reduce number of paragraphs in the abstract, e.g. p1 line 8 is a one sentence paragraph.

please check capital letters, e.g. Neural network (p3 line 26), neural network (p2 line 14) or Neural Network.

p3 line 22: (change . to ,) . . . of the networks, see Table 1 for selected Dates

Move p4 line 9-14 to line 6.

p4 line 27: introduce abbreviation GDAS (as written in line 30)

p4 line 28: add "the": . . . and the PFAAST radiative transfer model. . .

p5 line 3 add: The "uppermost cloud" top layer. . .

p5 line 11: reformulate "much of what"

p5 line 19: introduce physical unit "B" (in the lines before)

p5 line 23: $B\_11$ "for" neighboring pixels -> $B\_11$ "of the" neighboring Pixels

p5 line 25: avoid brackets

p 15 line 7, add "and": BT for water vapour channels at 6.7 "and" 7.3 mue m

p15 line 9/10/11, remove "." at the end of first entry, e.g. BT differences ""

p5 line 29 and thereafter: don't write $CO_2$ with cursive letters

p 6 line 19: hidden layer "for" the neural network -> hidden layer "of" the neural network

p 6 line 17-32: reduce the number of paragraphs. Don't create one sentence paragraphs.

p 7 line 18: write Ciwv in cursive letters

Chapter 4 (Table 5) use same order of retrievals as in table 3 and 4 (e.g. NN-NWP first)

p 7 last line: N-VIIRS -> NN-VIIRS

consider to write NoCO2 (in MetImageNoCO2) not in cursive letters.

Figure 1 (p21): consider to have the figures in the same order as the algorithms are mentioned in table 3, 4, and 5.

p 8, line 34 and following: avoid the abbreviation NN-CTTH, e.g. change: that the NN-CTTH all have -> that all NN retrievals have . . .

Figure 7 (p27): Could you please add a color scale instead of describing it with words.

p 9, line 11 and thereafter: cloud heights -> cloud "top" heights, avoid NN-CTTH abbreviation (which one do you mean? all NN retrievals or NN-MetImage or another one?)

p 10, line 12: avoid NN-CTTH -> specify which retrieval you referring to

p 10, line 15: The NN CTTH retrievals all have better results for low, medium and high clouds . . . (Clouds don't show results. . .)

p 10 line 16 "neither of which could be applied for the AVHRR1 instrument" can be skipped

---

## Referee Comment (RC2) · Anonymous Referee #1 · 1 Mar 2018

This paper describes a new approach to retrieving cloud-top height using a neural network. It is an interesting report and gives us hope for improved retrievals. It will be more valuable if additional information is provided. It is much improved from the original submission. I realize that this is a first step, but a bit more analysis would provide the springboard for the next steps. This is an important paper, but too brief.

Abstract

"Nowcasting" should be "nowcasting"

Here and elsewhere: please spell out the acronyms the first time they are used (e.g., MODIS, AVHRR)

Sec. 2.2 and 2.3: Please indicate nadir or viewing angles of the CALIOP and CPR.

[Figure]

Sec. 3.2 pg. 4, 25: while the CO2 absorbing band is generally referred to as the 15-$\mu$m band, the MODIS channels are in the 13.3-14.4 $\mu$m range.

Sec 3.3.2: Were the clouds single-layered or both single and multi-layered? It is not clear here. Please indicate if you are training only for single layered clouds or training for the topmost layer. Is there a lower optical depth limit of the clouds detected in the CALIOP 1-km product?

Sec. 4 Are there biases in any of the results for both CALIOP and CloudSat? The mean absolute error does not tell us any tendencies one way or the other. Knowing biases is critical. While MAE is an interesting and informative variable, it gives us less information about variability, which the standard deviation of the differences (SDD) along with the bias would provide us, especially when added to the MAE. Additions of the bias should be included in the tables and discussed. If there is no bias, then the SDD would still provide useful additional information and place the results in the same context as many previously published comparison studies. Addition of biases may help the discussion.

Pg. 8, 14: What is the motivation for comparing with CloudSat? Is this a better reference? If so, why use CALIPSO? If not, why is it here? How were the matches made on the larger CPR footprint? Are there sampling differences between CALIOP and CPR? The CPR often misses the top portions of ice clouds and has difficulty detecting clouds with small particles. If the biases discussed earlier are known, the CPR information might be useful if the results are interpreted more in the discussion section. Also, what is the vertical resolution of CloudSat? Would that impact the differences?

Pg.8, 26: The plots are distributions of the differences. Bias is the average of those differences. Please correct.

Sec. 5 The discussion section is very thin. There is a paucity of what the results shown in the figures and table might mean. For example, what do the differences computed using two different references, CALIOP and CPR, tell us? All samples, except in polar

regions are taken in midday or near midnight for Aqua. Could there be any diurnal impacts of training only with this dataset? What happens if the neighbouring pixel is turned off in the training? The conclusions state that that is an important input. Can its impact be quantified to support that conclusion?

Pg. 9, 22: It seems that using matches with Terra will not help much in the nonpolar regions. Is this a realistic possibility given the orbital differences?

Pg. 9, 30: This section is where the further work on the sources of error (e.g., various cloud types) could be presented. It would help the discussion considerably.

Sec. 6. More analysis in the discussion section would help flesh out this section.

---

## Referee Comment (RC3) · Anonymous Referee #1 · 12 Mar 2018

Thanks for the explanation for not including the bias and SDD. This is precisely the kind of discussion that belongs in the paper. Without this explanation and discussion, it would appear to many readers that something is being hidden by the authors.

The obvious question to most interested parties, particularly those who are potential users of the data, is, "Is the cloud height retrieved with this method, on average, in the right location? If not, how far away from the right altitude is it?" That is essentially the question both reviewers have asked. If I am assimilating or verifying a model output, I will want to put the cloud in the correct layer. An MAE of 500 m can just as easily be produced by all positive or all negative differences and thus I might expect to be within 500 m of the correct height on average, but I will not know if it is plus or minus 500 or if I am always biased high or low. The distributions in the current figures help but are

not quantitative. If I look at other cloud height data sources and see that they tell me whether I should expect to be too low or too high on average, I might be more inclined to use one of their datasets. For example, Hamann et al. (AMT, 2014) summarized their differences in bias, stdv and rmsd. Straightforward. It is not the whole story as argued in the response, but an important part. and one most people can relate to. The reader is not well served when obvious statistics are excluded. An explanation for why the bias and SDD are not included has been provided to the reviewers, but not to the readers. There is a lot of good discussion and information in your explanation about the retrievals that are important to understand. For example, the breakdown of biases according to cloud height is very helpful. The differences in bias between CPR and CALIPSO follows from some of my other comments. I find the paper unacceptable without such basic statistics. I think that the paper should include all of it: bias, SDD, Skew, Median, and MAE. The discussion then should be directed at explaining what the best measure should be and why one is better than the other. Part of that is already done in the supplement.

---

## Editor Comment (EC1) · A. M. Sayer (Editor) · 12 Mar 2018

Dear authors,

This is with reference to your response to the author comment concerning acronyms (e.g. MODIS, AVHRR). While these are indeed the commonly-known names for these instruments, all acronyms should be defined at the point of first use in the abstract and body text of the manuscript.

Best wishes,

Andrew
* * *

---

## Author Comment (AC1) · 12 Mar 2018

**Reply to anonymous Referee 2 comments to *Neural network cloud top pressure and height for MODIS**

Nina Håkansson et al.

**1 Overall quality of the discussion paper ("general comments"):**

**1.1 Referee comment:**

In the paper a novel retrieval of cloud top pressure and height using neural networks is presented. The presented retrieval technique is state of the art and an accurate technique. To account for different availabilities of channels on different satellites, a few modification of the neural network are investigated revealing the information content of the different channels. The new algorithms are compared to two reference algorithms, the CTTH algorithm of the NWCSAF PPS-v2014 and the MODIS collection 6 L2 height product. Additionally the algorithms is compared to CALIOP and CPR measurements. The quality of the algorithm is evaluated in terms of the mean absolute error (MAE). The improved quality of the results is impressive. From my point of view, I would request for at least another quality measure like standard deviation (similar to Tables 6 and 7). In overall, good work!

**Reply:**

We thank Referee 2 for this positive comment, and for the other valuable comments that will help us improve the paper further.

Regarding adding standard deviation, which was also requested by Referee 1, we chose the MAE as evaluation metric, over bias and standard deviation of differences (SDD), for many good reasons. However, some of the good reasons became clear to us first when we where faced with the request to include them in the article. Most important is that including bias and SDD of the error distribution intuitively gives the reader the mental picture of a Gaussian error distribution, centred at the bias. However we are dealing with skewed and even bimodal distributions as shown in Figure 2 and the mean is not at the centre of the distribution; i.e. the bias is not located at the peak of the error distribution.

The overall standard deviation is much affected by the largest errors. Some large errors are expected due to the differences between the passive and active sensors and the different FOV (field of view). Therefore we argue that the MAE is a better measure of variation of the error compared to SDD. The largest errors are of course also interesting, but when investigating these some care should be made to separate true errors from expected differences due to for example cloud edges in the FOV. We will consider including the Median absolute deviation (MAD) or the interquartile range (IQR) as these are more robust measures of variablity less sensitve to outliers compared to SDD.

See also the more detailed reply to comment 2.6 in the reply to Referee 1 regarding inclusion of bias and SDD. As the bias and SDD are traditionally used when evaluating cloud top height retrieval algorithms we will motivated in the article why they are not included.

**2 Individual scientific questions/issues ("specific comments")**

**2.1 Referee comment:**

p1 line 23: CTH might also be used in data assimilation of atmospheric motion vectors.

**Reply:**

We will consider adding this to the text.

**2.2 Referee comment:**

Introduction: A short description of the traditional technics to retrieve cloud top pressure and height could be added to the introduction. (or cite an overview paper like Hamann et al. "Remote sensing of cloud top pressure/height from SEVIRI: analysis of ten current retrieval algorithms." Atmospheric Measurement Techniques 7.9 (2014): 2839-2867.)

**Reply:**

We will add the suggested reference.

**2.3 Referee comment:**

The introduction should motivate, why it is expected that using machine learning, in particular neural networks, could improve the expected results.

**Reply:**

Many CTH retrieval algorithms including MODIS-C6 and PPS-v2014 include some fitting of temperatures to NWP temperature profiles. This is most difficult in the case of inversions both as one temperature occurs at several pressure heights in the profile but also as the inversions are often not captured accurately enough in the NWP temperature profile. Many different techniques are used to deal with this, for example PPS-v2014 will place the cloud at the inversion height if the temperature is not more than 0.5 to 2K lower than the temperature at the inversion. MODIS-C6 has another approach using climatological lapse rates over sea for clouds likely to be low. These kinds of fitting techniques a statistical machine learning technique could probably do better.

**2.4 Referee comment:**

Merge chapter 2.1.1 into chapter 3.2. (and skip the sentence (p3 line 10) "The MODIS Collection 6 cloud product were used as an independent. . .", you said that before).

**Reply:**

We will remove section 2.1.1 and move the information from it to section 2.1 and 3.2.

**2.5 Referee comment:**

Chapter 2.2 Add a short sentence, why you chose the CALIOP 1km product and not also 5km or 10km product which are more sensitive to optically thin clouds.

**Reply:**

The 1km CALIOP product was selected because it has the resolution closest to the MODIS resolution. It is expected that the thinnest cloud seen by CALIOP lidar is invisible to the passive imagers, so it should not be a problem that the thinnest clouds are missing in the 1km data. However we have also done some tests using AVHRR-GAC data and CALIOP 5km (Version 4) resolution for training (this is outside the scope of this article). The first tests show that results improve if the thinnest (0.05 or 0.1 in optical depth) clouds are excluded from the training. If these networks (trained on AVHRR-GAC) are applied on the validation data (MODIS) of this article the MAEs of the retrievals are between 76hPa to 79hPa. We will add a sentence about why 1km data was chosen.

**2.6 Referee comment:**

Chapter 2.4 add the version number of the ECMWF model and add product name and version number of the OSISAF data used in this study.

**Reply:**

We will do that.

**2.7 Referee comment:**

You might consider to add the PPS-v2014 and MODIS C6 algorithm to table 3 and 4.

**Reply:**

We will add them to Table 4, this will give the clear view of what channels are used for which method also for them. We suggest that they are not included in Table 3 as it describes *Network specific variables*. There is also an error in table 4, the NN-OPAQUE uses channel 12μm as described in Table 3 not channel 11μm. We will correct this as well.

**2.8 Referee comment:**

Please make the order of algorithms in table 3, 4 and 5 consistent.

**Reply:**

We did this in the first revision, and can not find any remaning inconsistencies.

**2.9 Referee comment:**

p5 line 5: how often is a pressure lower than 70 hPa retrieved?

**Reply:**

It varies with each network from 0 up to 0.05%. We will report this in the text.

**2.10 Referee comment:**

p5 line 10: Why did you choose this number of levels? Is it sufficient to use 6 levels to represent the boundary layer inversions or other small scale features?

**Reply:**

Five of the levels (surface, 950, 850, 700, 500) where already used in the PPS-v2014, so this was our starting point. We tested to use the troposphere pressure, but then the networks became very sensitive to the type of NWP-data used. Instead we added the 250hPa level to have one more high level. We did tested increasing the number of levels near the ground by adding levels at 800, 900 and 1000hPa, but the improvement was not large enough to motivate the extra computational time. One common problem for cloud height retrieval algorithms is that inversions are not represented accurately enough in the NWP data. As mentioned previously, MODIS-C6 instead uses climatological lapse-rates over sea to avoid this problem, other algorithms use sharpening techniques at the inversion. So it is not clear that more levels which would better represent the inversions in the NWP data would improve the neural network results, but this could be further investigated.

**2.11 Referee comment:**

p5 line 21: do you skip non cloudy pixels in the 5x5 pixel standard deviations?

**Reply:**

No, all pixels are included.

**2.12 Referee comment:**

p5 line 20: the $B_3.7$ has a solar component. Did you correct for this during day/night?

**Reply:**

No correction was made, that channel is used just like the others.

**2.13   Referee comment:**

p6 chapter 3.3.2: You chose to use specific days for training and others for validation. Given that you only use a limited number of days, wouldn't it be more to randomly select independent pixels from all available dates for training, validation, and testing to represent a larger variety of weather situations?

**Reply:**

Unfortunately all pixels in the dataset we have are not independent. A typical cloud is much larger than one pixel; there could be hundreds of pixels with almost identical data in the datasets. If we would randomly select independent pixels from all days for each dataset we would in practice use the same data all the time. This would cause the neural network to overtrain as the *during training validation data* would be the same as the *training data*. And in the last validation step results would be overly positive as also the *validation data* would be in practice the same as the *training data*.

Random sampling from all data has been suggested to us previously. We will therefore add the discussion to the article why it is not possible.

**2.14   Referee comment:**

p 6 line 19: Did you test other configurations that 30/15 neurons in the first/second layer? If yes, how was the performance?

**Reply:**

We did one test with 20/15 this network (NN-AVHRR) was 1 hPa worse and one with 30/45/45 (NN-AVHRR) this was 2.5hPa better but also took 5 times as long time to retrieve pressure.

**2.15   Referee comment:**

Chapter 3.3.3: May batch size and momentum be changed during the training process?

**Reply:**

No they can't.

**2.16   Referee comment:**

 consider to discuss the solar component of the 3.7 mue m channel. To my opinion this NN could perform better when corrected for that (e.g. adding the solar time as input variable).

**Reply:**

We considered adding the sun zenith angle as variable; however we can not decide how the neural network would use it. In the data we do not have all sun zenith angles present globally. It could be that the neural network would use the sun zenith angle to decide that during this time of day clouds of a particular height are most common. It does not have to be bad though and can be tested in future studies. The performance of NN-AVHRR1 could probably be improved it the solar component of 3.7 is treated explicitly. We will consider to add the discussion.

**2.17 Referee comment:**

p8 line 5: Maybe express it positively: All NN can reproduce a clear bi-modal pdf very similar to CALIPSO, the pdf of PPS-v2014 deviates from this shape . . .

**Reply:**

We will change the formulation, thank you for the suggestion.

**2.18 Referee comment:**

p8 line 7: It is written "for the best performing network". Did you train several networks for one channel configuration? If so, could you describe the number of trained networks in chapter 3.2.2, please?

**Reply:**

With the *the best performing networks* we meant that the NN-NWP, NN-OPAQUE, NN-BASIC and NN-BASIC-CIWV was excluded. We will clarify this in the text.

**2.19 Referee comment:**

p 8, line 11: according to my table 6, the NN-MetImage is better than the NN-MetImage- NoCO2.

**Reply:**

Yes it does! However the NN-MetImage does not perform well for higher satellite zenith angles. Only networks that perform well for all satellite zenith angles are discussed in this sentence. We will consider to reformulate to make it clearer.

**2.20 Referee comment:**

p 8, line 15: do you have an idea, why the MAE against CPR is larger than the MAE against CALIOP for NN-MetImage and NN-MetImage-NoCO2?

**Reply:**

This we think it partly because the NN-MetImage and NN-MetImage-NoCO$_2$ have some skill in predicting very thin high clouds that are not detected by the CPR-Radar. We will add discussion about this.

**2.21 Referee comment:**

p 9, line 9: could you please describe a bit more in detail the differences seen in Figure 7?

**Reply:**

We will describe the differences in more detail. The blue squares for PPS-v2014 (c) are due to the temperature retrieval for 32x32 pixels in one go. We can see that a lot of high clouds are by NN-AVHRR placed higher (pixels that are blue in (c), are white in (a)). For NN-AVHRR in (a) we can see that the large area with low clouds in the lower left corner get a consistent cloud height (the same orange colour everywhere).

**2.22 Referee comment:**

Chapter 5 Discussion: Could you also comment on applying your NN technique on geostationary satellites? What would be the main differences/challenges?

**Reply:**

This technique should not be limited to polar orbiting satellites. As the instrument SEVIRI has the two most important channels at 11μm and 12μm it should be possible to apply the technique to SEVIRI data. More data (in terms of number of days) compared to MODIS may be needed to produce enough matches. As the SEVIRI resolution is coarser results might be degraded compared to MODIS. Matches of SEVIRI with CALIOP will occur at many different satellite zenith angles. This might make it possible to use the CO$_2$ channel on SEVIRI to improve results without losing performance skill at high satellite zenith angles. We will comment on using the NN-CTTH technique for geostationary satellites in the paper.

**A compact listing of purely technical corrections ("technical corrections": typing errors, etc.)**

Replies to a few of the technical corrections. For the ones not mentioned here we will follow the suggestions from Referee 2.

**2.23 Referee comment:**

- p5 line 29 and thereafter: don't write CO2 with cursive letters

– consider to write NoCO2 (in MetImageNoCO2) not in cursive letters.

**Reply:**

We will keep the notation with subscript but without cursive letters.

**2.24   Referee comment:**

Figure 1 (p21): consider to have the figures in the same order as the algorithms are mentioned in table 3, 4, and 5.

**Reply:**

This is a reasonable request. However it is also nice to have the two AVHRR based algorithms next to each other so they can be compared. Also this order makes the two networks performing bad at high satellite zenith angles appear on the last row. And changing the order increases the risk to mix them up in later references. If someone refers to the bad satellite zenith angle behaviour in Figure 2 (h) in the discussion paper, and an interested reader by accident finds the final revised paper (assuming there will be one) and finds the result for NN-AVHRR1 in that sub figure that is not good. As there are also good reasons to keep the current order of sub figures we argue that ther order should not be changed.

**2.25   Referee comment:**

– p 8, line 34 and following: avoid the abbreviation NN-CTTH, e.g. change: that the NN-CTTH all have -> that all NN retrievals have . . .

– avoid NN-CTTH abbreviation (which one do you mean? all NN retrievals or NN-MetImage or another one?)

– p 10, line 12: avoid NN-CTTH -> specify which retrieval you referring to

**Reply:**

We will choose to keep the NN-CTTH as the name for the neural network method in the paper. We will better present it as the name and avoid using it where it might be confusing.

**2.26   Referee comment**

p 10 line 16 "neither of which could be applied for the AVHRR1 instrument" can be skipped

**Reply:**

We will consider to reformulate. For the processing of longer climate data records it can be of interest to know if an algorithm can also be applied for AVHRR1.

---

## Author Comment (AC2) · 12 Mar 2018

**Reply to anonymous Referee 1 comments to *Neural network cloud top pressure and height for MODIS**

Nina Håkansson et al.

**1 General comment**

**1.1 Referee comment:**

This paper describes a new approach to retrieving cloud-top height using a neural network. It is an interesting report and gives us hope for improved retrievals. It will be more valuable if additional information is provided. It is much improved from the original submission. I realize that this is a first step, but a bit more analysis would provide the springboard for the next steps. This is an important paper, but too brief.

**Reply:**

We thank Referee 1 for acknowledging the paper as important and for all interesting comments that will help us extend the analysis of the paper.

**2 Specific comments**

**2.1 Referee comment:**

"Nowcasting" should be "nowcasting"

**Reply:**

We will correct this.

**2.2 Referee comment:**

Here and elsewhere: please spell out the acronyms the first time they are used (e.g., MODIS, AVHRR)

**Reply:**

As both acronyms MODIS and AVHRR are better known than their written-out form we argue to keep just the acronym for these two, following the manuscript-preparation guidelines. However we will check all acronyms again as there are others (some noted also by Referee 2) that we should define.

**2.3 Referee comment:**

Sec. 2.2 and 2.3: Please indicate nadir or viewing angles of the CALIOP and CPR.

**Reply:**

We will add that the viewing angle for CALIOP is $3°$, and for CPR $0.16°$. In Section 2.1 we will also add information of the satellite zenith angles for the MODIS data. For the matches with CPR the MODIS satellite zenith angle varies between $0.04°$ and $19.26°$; and for matches with CALIOP between $0.04°$ and $19.08°$.

**Reply:**

**2.4 Referee comment:**

Sec. 3.2 pg. 4, 25: while the CO2 absorbing band is generally referred to as the $15 - \mu m$ band, the MODIS channels are in the $13.3 - 14.4\mu m$ range.

**Reply:**

We will correct the channel ranges mentioned.

**2.5 Referee comment:**

Sec 3.3.2: Were the clouds single-layered or both single and multi-layered? It is not clear here. Please indicate if you are training only for single layered clouds or training for the topmost layer. Is there a lower optical depth limit of the clouds detected in the CALIOP 1-km product?

**Reply:**

Because it is currently not clear enough we will explicitly state that both single and multilayer clouds where included. We will also clarify that we used the uppermost layer of the top layer pressure variable as this is missing in the text (also noted by Referee 2).

Clouds optically thick enough to be detected when averaging the lidar data on 1km resolution should be included in the CALIOP 1km data. As we actually have the total optical depth from the 5km included in our match-up data (needed for other studies) we checked the lowest reported optical depth in 5km data for clouds that are detected in the 1km data, it was 1.5e-05.

**2.6  Referee comment:**

Sec. 4 Are there biases in any of the results for both CALIOP and CloudSat? The mean absolute error does not tell us any tendencies one way or the other. Knowing biases is critical. While MAE is an interesting and informative variable, it gives us less information about variability, which the standard deviation of the differences (SDD) along with the bias would provide us, especially when added to the MAE. Additions of the bias should be included in the tables and discussed. If there is no bias, then the SDD would still provide useful additional information and place the results in the same context as many previously published comparison studies. Addition of biases may help the discussion.

**Reply:**

We do not agree that biases are at all critical, but rather we claim that there is a large risk of misinterpretation of the biases as we are handling non-Gaussian skewed distributions. Our preference for the MAE over bias and SDD has grown over the years, but first faced with the direct request to include also bias in this article did we fully investigate why they are less useful or even misleading. We thank Referee 1 for raising this question. And as bias and SDD are often included in previously published comparison studies the exclusion of them need to be motivated in the paper.

To make our point clear, tables with biases and SDD are included in this answer for CALIPSO (Table A1), and CloudSat (Table A2) but we argue that they should best be left out of the article.

Consider bias and SDD for low level clouds for PPS-v2014 and NN-AVHRR in Table A2, seeing these results the PPS-v2014 must be the better algorithm for low level clouds. The small improvement in SDD with 16m can surely not be worth the 233m higher bias! However considering the distribution on the differences (Figure 2 (e)) in combination with the better MAE in Table 7 we would claim that the NN-AVHRR is the better algorithm also for low level clouds. How could the bias and SDD indicate the opposite, what is going on here? The low bias for PPS-v2014 is because it underestimates the height for most low clouds which nicely compensates for amount of clouds being places 1.5-2km too high. The 472m bias for NN-AVHRR indicates a difference distribution for NN-AVHRR centred at 472m but that again is not reflected in Figure 2 (e). The peak of the difference distribution seams to be located much closer to zero. The NN-AVHRR distribution in fact places 89.3% of the low clouds within +/-1.5km, to compare with 83% PPS-v2014. And considering this part (most) of data the bias for NN-AVHRR is 19m, to compare with -214m for PPS-v2014. So the large positive bias of 472m does not mean that the difference distribution peaks at 472m it is instead a consequence of the fact that the error distribution is skewed; low level clouds are more often placed too high than too low. This naturally is the case as there is always fixed limit (ground) for how low clouds could realistically be placed. If we consider bias and SDD also the skewness of the distributions should be considered at the same time, but this makes interpretation of the results even harder.

Regarding SDD we argue that MAE already provides a better estimate of the spread of the data. The MAE is not dependent of the bias and it is less influenced by outliers and the largest errors compared to the SDD. Focusing on SDD, during algorithm development, could make it seem much more important to improve pixels that are 15km off to just 10km off compared to improve pixels that are 2km off. Considering again Figure 2 (e), how could there be such a small difference in SDD in Table A2

between NN-AVHRR and PPS-v2014, is that not unrealistic? Surely the NN-AVHRR distribution seems more centred around zero? If we look at the cases with difference larger than 4km which are not visible in Figure 2 (e) for NN-AVHRR they have a bias of 6.7km and (and consist of 4.03% of the low cloud data) and for PPS-v2014 they will have a bias of 6.3km and (consist of 3.7% of the low clouds data). This is what makes the SDD improvement so small. For the data with bias below 4km (more than 95% of the data) the SDD for NN-AVHRR it is 815m, compared to 1029m for PPS-v2014. As the SDD puts more focus on the largest errors, the increase of 0.3% low level clouds being predicted much (>4km) too high and the 0.4km increase in absolute error for these clouds evens out the improvement seen in the error distribution for the majority (96%) of the data (Figure 2 (e)).

Of course the worst cases are important. The increased amount (0.3%) of absoulte errors above 4km, discussed in previous section, could be a true degradation of the performance for NN-AVHRR compared to PPS-v2014. However we should specially remember that we are handling different FOVs and different instruments and we do expect differences for some pixels for example at cloud edges. When focusing on the worst performing part of data, neighbouring pixels both for the imager and the radar (or lidar) should be considered to decide if we are at a cloud edge. When investigating the worst cases it is important to make an effort to separate the true failures of the algorithms from errors expected from instrument and field of view differences.

To make clear that there is no general increase of outliers for the neural network method we made the same analysis as above and calculated amount of data with large (>4km) absolute errors for all cases in Table 6 and Table 7. In the example discussed above, PPS-v2014 low level clouds validated with CloudSat, is the only case where the neural networks have a larger percentage of large errors compared to PPS-v2014. MODIS-C6 data have a greater amount of large errors compared to all of the neural networks in all cases.

In addition to the other problems with interpreting bias for cloud top height retrieval, the overall bias are very dependent on the frequency distribution of high, medium and low clouds which is dependent on the validation dataset used and the performance of the cloud mask algorithm. This means that even if bias and SDD are included, the total bias and SDD should really not be compared between different studies that use different validation datasets and different cloud masks. The possibility to compare statistical measures between studies is further limited by the usage of different validation strategies including: filtering, averaging, only single layers, only low clouds etc.

As another illustration of the problem with bias consider a fictional cloud height retrieval algorithm with an error distribution that are Gaussian distributed ($\sim N(0, 2km)$). The zero bias here actually marks the centre of the distribution and together with the SDD describes the expected errors well. Now an improvement to the algorithm is made and clouds that where before placed more than 2km too low are now placed exactly at the truth. This improvement effects one of the tails and the largest part of the error distribution stays the same, although we now have quite a high percentage of data with zero error. Our mental picture with the error centred at zero is still valid (and the mode and the median are still 0). However if we calculate the bias for this truncated (and therefore skewed distribution) it is now 480m. A bias of 480 meters indicates an algorithm that generally places clouds too high. But as we know the underlying distribution we know that is not true; the algorithm just never places any clouds much too low.

It is also possible for a distribution to have a bias close to zero and not be exceptionally good. See for example the NN-AVHRR1 bias of only -8m in Table A2, this stands out as clearly better than all the others. However considering the MAE in Table 7 and the median in Table A4 it is clear that also this network performs in line with the others.

We agree that information about tendencies are lacking in the tables in the result section, the MAE does not provide information if clouds are generally over or under estimated. Figure 2 provides a view of the error distribution for the three algorithms (although only one of the neural networks is included). Here also the sometimes bimodal behaviour is evident (for low level clouds PPS-v2014, and high level clouds MODIS-C6) and the skewness of the distributions are evident. We will extend the discussion about the results in Figure 2. We will add a tendency measure, the median, which is more suitable for skewed distributions to Table 6 and Table 7 and we thank Referee 1 for suggesting tendency measures as an improvement to the article. We will also add a discussion why MAE and median are used and not the traditional bias and SDD. We will mention that the median should be interpreted with caution and that it will not be the same as the mode (where the peak of the error distribution is) again because the distributions are skewed and in some cases even bimodal. We suggest that we add Figure A1 to the article to help the discussion by visually showing why numercial valuse for the biases where excluded. In Figure A1 the error distributions are plotted together with Gaussian distributions with the same bias and SDD and the differences are evident. For the completeness we here included the same figure for low, medium and high (Figure A2-A4) cloud classes however we suggest that these will be left out of the article. These could be included as supplementary figures. We will also consider including the Median absolute deviation (MAD) or the interquartile range (IQR) as these are more robust measures of variablity less sensitve to outliers compared to SDD.

To summarize the above discussion: we are dealing with skewed, non-normal and even bimodal distributions and this makes the median a better measure of tendency than the bias. As we are not very interested in how large the largest errors are, especially as these are partly expected due to the sensor differences, the MAE is a better measure of spread compared to SDD.

See also further comments in the reply to Referee 2 who also wanted SDD included.

**2.7 Referee comment:**

Pg. 8, 14: What is the motivation for comparing with CloudSat? Is this a better reference? If so, why use CALIPSO? If not, why is it here? How were the matches made on the larger CPR footprint? Are there sampling differences between CALIOP and CPR? The CPR often misses the top portions of ice clouds and has difficulty detecting clouds with small particles. If the biases discussed earlier are known, the CPR information might be useful if the results are interpreted more in the discussion section. Also, what is the vertical resolution of CloudSat? Would that impact the differences?

**Reply:**

The CloudSat validation are included to get an independent source of validation, not better just different. We will improve the discussion regarding this, see reply to comment 2.9. Nearest neighbour matching is used; we will add this information in the article as the description of the matching method is now missing.

Clouds not detected at all by CloudSat are not a problem as it simply means that we will have less data. That the CPR often misses the top portions could partly explain why results are not improving for NN-MetImage and NN-MetImage-NoCO$_2$ (compared to NN-MERSI-2) when validating with CloudSat. We will add this in the discussion.

The vertical resolution of CPR is 0.5km this means that we should expect MAE higher than 250m. We will add discussion about this and about the medians that will be added to the result section instead of biases.

**2.8 Referee comment:**

Pg.8, 26: The plots are distributions of the differences. Bias is the average of those differences. Please correct.

**Reply:**

We will correct that.

**2.9 Referee comment:**

Sec. 5 The discussion section is very thin. There is a paucity of what the results shown in the figures and table might mean. For example, what do the differences computed using two different references, CALIOP and CPR, tell us? All samples, except in polar regions are taken in midday or near midnight for Aqua. Could there be any diurnal impacts of training only with this dataset? What happens if the neighbouring pixel is turned off in the training? The conclusions state that that is an important input. Can its impact be quantified to support that conclusion?

**Reply:**

We thank Referee 1 for the suggestions and valuable comments that will help to improve the discussion section.

The usage of two validation truths strengthens our results. The CloudSat results confirm that the improvements are not only due to that the neural networks have learnt to replicate errors of CALIOP. (For the argumentation let us pretend that CALIOP would always place clouds at 5km height if the surface pressure is 1000 hPa, a neural network could learn this but it would not really improve the accuracy of the retrieved cloud top height). Considering the large improvement it was not an alarming risk that the neural network was learning only to mimic CALIOP errors, but with the independent validation truth CloudSat this is confirmed. We will better motivate the inclusion of CloudSat in the paper.

What happens if the neighbouring pixels are not used is to some extent seen in Table 5, but not well enough described in the results (page 7, line 19) and overlooked in the discussion section. We will discuss these results in more detail in the discussion section to support better the statement in the conclusion.

There might be diurnal impact not captured in the current dataset. However results are valid for Aqua which we trained for. Applying similar neural networks to other sensors with different filter functions and ECT will require additional work or validation not in the scope of this paper.

**2.10 Referee comment:**

Pg. 9, 22: It seems that using matches with Terra will not help much in the non-polar regions. Is this a realistic possibility given the orbital differences?

**Reply:**

As latitude is not used as a variable, data for higher satellite zenith angles included for Polar regions could help also in non-Polar regions. However it might be that the high latitude matches will not help the network the if varity of weather situations and cloud heights at high latitudes are too small. This must be tested. We will extend discussion regarding adding Terra matches.

**2.11 Referee comment:**

Pg. 9, 30: This section is where the futher work on the sources of error (e.g., various cloud types) could be presented. It would help the discussion considerably.

**Reply:**

We will extend the discussion section with help of the questions raised by the Referees.

**2.12 Referee comment:**

Sec. 6. More analysis in the discussion section would help flesh out this section.

**Reply:**

We will extend Section 6, reflecting what is added to Section 5.

**Table A1.** Bias, standard deviation (SDD) and skewness of the error distribution in meters for different algorithms compared to CALIOP top layer altitude. Skewness where calcualted with the scipy.stats skew function. Running the skewtest function show that all distribution have a skewness that differ from the normal distribution. Interpret with caution; as the distributions are skewed the bias are not at the center of the distribution.

| | Bias [m] | | | | Std [m] | | | | Skewness | | | |
|---|---|---|---|---|---|---|---|---|---|---|---|---|
| | all | low | medium | high | all | low | medium | high | all | low | medium | high |
| PPS-v2014 | -1473 | 295 | -373 | -3436 | 2797 | 1409 | 1611 | 2799 | -1.0 | 3.0 | 0.2 | -0.9 |
| MODIS-C6 | -1162 | 204 | -730 | -2524 | 2854 | 1518 | 2179 | 3313 | -1.5 | 2.7 | 0.6 | -1.5 |
| NN-AVHRR | -416 | 427 | 362 | -1447 | 2175 | 973 | 1330 | 2691 | -1.8 | 3.9 | 0.7 | -1.4 |
| NN-VIIRS | -422 | 369 | 199 | -1350 | 2049 | 941 | 1194 | 2565 | -2.0 | 4.6 | 0.5 | -1.6 |
| NN-MERSI-2 | -431 | 342 | 123 | -1318 | 1967 | 930 | 1080 | 2466 | -2.1 | 4.7 | 0.7 | -1.7 |
| NN-MetImage-NoCO$_2$ | -428 | 279 | 63 | -1234 | 1938 | 955 | 1128 | 2448 | -2.1 | 4.8 | 0.8 | -1.8 |
| NN-MetImage | -306 | 242 | 12 | -907 | 1774 | 979 | 1063 | 2271 | -2.1 | 5.4 | 0.7 | -2.0 |
| NN-AVHRR1 | -593 | 374 | 193 | -1736 | 2260 | 931 | 1360 | 2771 | -1.8 | 3.7 | 0.4 | -1.3 |

**Table A2.** Bias, standard deviation (SDD) and skewness of the error distribution in meters for different algorithms compared to CPR (Cloud-Sat) Height. Skewness where calcualted with the scipy.stats skew function. Running the skewtest function show that all distribution have a skewness that differ from the normal distribution. Interpret with caution; as the distributions are skewed the bias are not at the center of the distribution.

| | Bias [m] | | | | Std [m] | | | | Skewness | | | |
|---|---|---|---|---|---|---|---|---|---|---|---|---|
| | all | low | medium | high | all | low | medium | high | all | low | medium | high |
| PPS-v2014 | -1122 | 239 | -509 | -2056 | 2179 | 1605 | 1859 | 2097 | -0.1 | 2.7 | 0.5 | -0.5 |
| MODIS-C6 | -630 | 445 | -528 | -1217 | 2555 | 2123 | 2481 | 2602 | -0.1 | 2.9 | 0.9 | -1.2 |
| NN-AVHRR | 151 | 472 | 411 | -114 | 1945 | 1589 | 1774 | 2128 | 0.2 | 3.8 | 1.2 | -0.6 |
| NN-VIIRS | 142 | 459 | 314 | -85 | 1918 | 1687 | 1765 | 2052 | 0.5 | 4.2 | 1.5 | -0.7 |
| NN-MERSI-2 | 139 | 429 | 263 | -55 | 1807 | 1651 | 1684 | 1903 | 0.5 | 4.0 | 1.7 | -0.9 |
| NN-MetImage-NoCO$_2$ | 149 | 352 | 204 | 25 | 1849 | 1689 | 1742 | 1955 | 0.7 | 4.1 | 1.8 | -0.7 |
| NN-MetImage | 290 | 397 | 228 | 260 | 1867 | 1831 | 1824 | 1899 | 0.9 | 4.0 | 1.9 | -0.8 |
| NN-AVHRR1 | -8 | 520 | 303 | -400 | 1974 | 1651 | 1752 | 2117 | 0.0 | 3.5 | 1.0 | -0.8 |

**Table A3.** Median error in meters for different algorithms compared to CALIOP top layer altitude. The final validation dataset, containing 1793142 pixels (45% high, 39% low and 16% medium level clouds), where all algorithms had a cloud height is used. The low, medium and high classes are from CALIOP feature classification flag. A small amount 0.2% of the pixels were excluded because of missing height or pressure below 70hPa for any of the algorithms. Interpret with caution; it is not the same as the mode of the error distribution. These columns is suggested to be included in Table 6.

| | Median [m] | | | |
| --- | --- | --- | --- | --- |
| | all | low | medium | high |
| PPS-v2014 | -653 | -50 | -77 | -2923 |
| MODIS-C6 | -620 | -19 | -666 | -1588 |
| NN-AVHRR | 47 | 211 | 291 | -797 |
| NN-VIIRS | 25 | 179 | 177 | -719 |
| NN-MERSI-2 | -6 | 156 | 75 | -706 |
| NN-MetImage-NoCo$_2$ | -53 | 94 | 20 | -607 |
| NN-MetImage | -20 | 71 | -3 | -359 |
| NN-AVHRR1 | -46 | 193 | 177 | -1101 |

**Table A4.** Median error in meters for different algorithms compared to CPR (CloudSat) Height. The final validation dataset, containing 1121199 pixels (53% high, 27% low and 21% medium level clouds) is used. The low, medium and high classes are derived comparing the CloudSat height to the NWP height at 440hPa and 680hPa. A cloudy threshold of 30% is used for CloudSat. Interpret with caution; it is not the same as the mode of the error distribution. These columns is suggested to be included in Table 7.

| | Median [m] | | | |
| --- | --- | --- | --- | --- |
| | all | low | medium | high |
| PPS-v2014 | -849 | -156 | -298 | -1797 |
| MODIS-C6 | -384 | 49 | -598 | -619 |
| NN-AVHRR | 94 | 25 | 213 | 141 |
| NN-VIIRS | 75 | 3 | 120 | 165 |
| NN-MERSI-2 | 46 | -24 | 17 | 177 |
| NN-MetImage-NoCo$_2$ | 13 | -101 | -43 | 216 |
| NN-MetImage | 91 | -97 | -41 | 459 |
| NN-AVHRR1 | 36 | 54 | 151 | -57 |

[Figure]

**Figure A1.** Error distribution compared to CPR (CloudSat) (left) and CALIOP (right) with biases and medians marked. In grey the Gaussian distribution with the same bias and standard deviation as the error distribution is shown. The percent of data is calculated in 0.1km bins. All height classes (low, medium and high) are included. Note that the values on the y-axis are dependent of the bin size. The peak at 6% for NN-AVHRR in subplot (f), means that 6% of the retrieved heights are between the CALIOP height and the CALIOP height + 0.1km. This figure is suggested to be included in the main article. The x-axis is cut at 4.5km and some part of the ditribution is not shown. For PPS-v2014 92% is visible for CloudSat and 86% is visible for CALIPSO. The corresponding numbers are for MODIS-C6 89% and 90% and for NN-AVHRR 94% and 96%.

[Figure]

**Figure A2.** Error distribution compared to CPR (CloudSat) (left) and CALIOP (right) with biases and medians marked. The percent of data is calculated in 0.1km bins. Only the CALIOP low class is included. In grey the Gaussian distribution with the same bias and standard deviation as the error distribution is shown.

[Figure]

**Figure A3.** Error distribution compared to CPR (CloudSat) (left) and CALIOP (right) with biases and medians marked. The percent of data is calculated in 0.1km bins. Only the CALIOP medium class is included. In grey the Gaussian distribution with the same bias and standard deviation as the error distribution is shown.

[Figure]

**Figure A4.** Error distribution compared to CPR (CloudSat) (left) and CALIOP (right) with biases and medians marked. The percent of data is calculated in 0.1km bins. Only the CALIOP high class is included. In grey the Gaussian distribution with the same bias and standard deviation as the error distribution is shown.

---

## Author Comment (AC3) · 15 Mar 2018

We will follow the suggestion from Referee 1 and include all measures along with the discussion in the paper.

Best Regards

Nina Håkansson
* * *

---

## Author Response (AR1)

**Reply to anonymous Referee 1 comments to** *Neural network cloud top pressure and height for MODIS**

Nina Håkansson et al.

**1 General comment**

**1.1 Referee comment:**

This paper describes a new approach to retrieving cloud-top height using a neural network. It is an interesting report and gives us hope for improved retrievals. It will be more valuable if additional information is provided. It is much improved from the original submission. I realize that this is a first step, but a bit more analysis would provide the springboard for the next steps. This is an important paper, but too brief.

**Reply:**

We thank Referee 1 for acknowledging the paper as important and for all interesting comments that will help us extend the analysis of the paper.

**2 Specific comments**

**2.1 Referee comment:**

"Nowcasting" should be "nowcasting"

**Reply:**

We have correct this.

**Changes in manuscript:**

- page 1 line: 2
- page 2 line: 27

**2.2 Referee comment:**

Here and elsewhere: please spell out the acronyms the first time they are used (e.g., MODIS, AVHRR)

**Reply:**

We have correctes this. We had misinterpreted manuscript-preparation guidelines regarding AVHRR and MODIS. We have also updated the manuscript to use the correct acronym CPR (CloudSat) (Cloud Profiling Radar for CloudSat (CLOUD SATellite)) everywhere.

**Changes in manuscript:**

- page 1 line: 3-5, 7-12
- page 2 line: 7, 16-23, 27-28, 31-33
- page 3 line: 11-12, 17, 20
- page 4 line: 4, 15-16
- page 5 line: 31-33
- page 7 line: 5-6
- CloudSat changed to CPR (CloudSat) many occurances

**2.3 Referee comment:**

Sec. 2.2 and 2.3: Please indicate nadir or viewing angles of the CALIOP and CPR.

**Reply:**

We have added that the viewing angle for CALIOP is  $3^{\circ}$ , and for CPR  $0.16^{\circ}$ . In Section 2.1 we have also added information of the satellite zenith angles for the MODIS data. For the matches with CPR the MODIS satellite zenith angle varies between  $0.04^{\circ}$  and  $19.26^{\circ}$ ; and for matches with CALIOP between  $0.04^{\circ}$  and  $19.08^{\circ}$ .

**Changes in manuscript:**

- page 4 line: 4-5, 10, 17
- page 9 line 31-32 made clear what was meant by near NADIR observations.

**Reply:**

**2.4 Referee comment:**

Sec. 3.2 pg. 4, 25: while the CO2 absorbing band is generally referred to as the  $15 - \mu m$  band, the MODIS channels are in the  $13.3 - 14.4\mu m$  range.

**Reply:**

We have corrected the channel ranges mentioned.

**Changes in manuscript:**

- page 5 line: 29

**2.5 Referee comment:**

Sec 3.3.2: Were the clouds single-layered or both single and multi-layered? It is not clear here. Please indicate if you are training only for single layered clouds or training for the topmost layer. Is there a lower optical depth limit of the clouds detected in the CALIOP 1-km product?

**Reply:**

To make it clearer we have explicitly stated that both single and multilayer clouds where included. We have also clarified that we used the uppermost layer of the top layer pressure variable as this is missing in the text (also noted by Referee 2).

**Changes in manuscript:**

- page 6 line: 6-7

Clouds optically thick enough to be detected when averaging the lidar data on 1km resolution should be included in the CALIOP 1km data. As we actually have the total optical depth from the 5km included in our match-up data (needed for other studies) we checked the lowest reported optical depth in 5km data for clouds that are detected in the 1km data, it was 1.5e-05.

**2.6 Referee comment:**

Sec. 4 Are there biases in any of the results for both CALIOP and CloudSat? The mean absolute error does not tell us any tendencies one way or the other. Knowing biases is critical. While MAE is an interesting and informative variable, it gives us less information about variability, which the standard deviation of the differences (SDD) along with the bias would provide us, especially when added to the MAE. Additions of the bias should be included in the tables and discussed. If there is no bias, then the SDD would still provide useful additional information and place the results in the same context as many previously published comparison studies. Addition of biases may help the discussion.

**2.6.1 Referee comment (RC3):**

Thanks for the explanation for not including the bias and SDD. This is precisely the kind of discussion that belongs in the paper. Without this explanation and discussion, it would appear to many readers that something is being hidden by the authors. The obvious question to most interested parties, particularly those who are potential users of the data, is, "Is the cloud height

retrieved with this method, on average, in the right location? If not, how far away from the right altitude is it?" That is essentially the question both reviewers have asked. If I am assimilating or verifying a model output, I will want to put the cloud in the correct layer. An MAE of 500 m can just as easily be produced by all positive or all negative differences and thus I might expect to be within 500 m of the correct height on average, but I will not know if it is plus or minus 500 or if I am always biased high or low. The distributions in the current figures help but are not quantitative. If I look at other cloud height data sources and see that they tell me whether I should expect to be too low or too high on average, I might be more inclined to use one of their datasets. For example, Hamann et al. (AMT, 2014) summarized their differences in bias, stdv and rmsd. Straightforward. It is not the whole story as argued in the response, but an important part. and one most people can relate to. The reader is not well served when obvious statistics are excluded. An explanation for why the bias and SDD are not included has been provided to the reviewers, but not to the readers. There is a lot of good discussion and information in your explanation about the retrievals that are important to understand. For example, the breakdown of biases according to cloud height is very helpful. The differences in bias between CPR and CALIPSO follows from some of my other comments. I find the paper unacceptable without such basic statistics. I think that the paper should include all of it: bias, SDD, Skew, Median, and MAE. The discussion then should be directed at explaining what the best measure should be and why one is better than the other. Part of that is already done in the supplement.

**Reply:**

In our first reply we argued that bias and SD should be excluded because the error ditributions are non-Gaussian. We where convinced by the arguments of Reviewer 1 that all the measures along with the discussion of them belong in the paper. Specially as the bias and SDD are something most people can relate to, and that the intuitive way is to interpret them as describing a Gaussian distribution.

Therefore the paper is updated with bias, SDD, Skew, median and to help the dicussion and interpretation of the statistics also IQR (interquartile range), RMSE (root mean square error) and mode where included. To give the potential users more quantitative information on the errors to expect and to help the discussion also percentage of absolut error above 0.25, 0.5, 1 and 2 km were included.

The extended discussion and result sections where combined into one section with several subsections: Validation with CALIOP top layer pressure, Discussion of statistics measures for non-Gaussian error distributions, Validation with CALIOP and CPR (CloudSat) height, Validation separated for low, medium and high level clouds, Validation with CALIOP separated for different cloudtypes, Geografical aspects of the NN-CTTH performance and Future work and challenges.

**Changes in manuscript:**

- Discussion of measures and result added to abstract at page 1 line: 17-24.
- Removed lines from abstract to keep it short at page 1 line 27 and page 2 line: 1-2.

- Moved lines to page 8 line 27-33 (from page 13 line 33-35 and page 14 line 1-4) as they make more sense here after the reorganization of the Result section.
- Reformulated (see answer to 2.21 to Reviwer 2) and moved lines to page 9 line 1-5 (from page 15 line 4-6) as it makes more sense here.
- This is most of the part of the old Results section conserning pressure is now found in Section 4.1 at page 9. Notice that lines 28-30 where moved to line 9-11.
- The new section with discussion of measures for non-Gaussian error distributions are added at page 10 and 11 Section 4.2.
- The old part of the Result section that were treating height is now in Section 4.3 and 4.4 page 12-14. Results of the additional measures (mode, median RMSE etc.) are included here.
- The geographical aspects of the NN-CTTH now as a separate section at page 14 Section 4.6.
- Removed page 15 line 8-9 due to section reorganization.
- Moved page 15 line 10-11 moved to conclusion page 17 line 14-16
- Moved page 15 line 13-15 to page 16 line 4-6.
- Moved one sentence from page 16 line 22-23 to page 17 line 16-18. Also reformulated it (See answer 2.9).
- Conclusions regarding the added masures included at page 16 line 29-33 and at page 17 line 1-2.
- Added conclusions related to non-Gaussian error distributions at page 17 line 8-13.
- Table 6 and Table 7 replaced with four new tables pages 27-32.
- The figure referenced in section 4.2 is added at page 41.

**2.7 Referee comment:**

Pg. 8, 14: What is the motivation for comparing with CloudSat? Is this a better reference? If so, why use CALIPSO? If not, why is it here? How were the matches made on the larger CPR footprint? Are there sampling differences between CALIOP and CPR? The CPR often misses the top portions of ice clouds and has difficulty detecting clouds with small particles. If the biases discussed earlier are known, the CPR information might be useful if the results are interpreted more in the discussion section. Also, what is the vertical resolution of CloudSat? Would that impact the differences?

**Reply:**

The CloudSat validation are included to get an independent source of validation, not better just different. We have improved the discussion regarding this, see also reply to comment 2.9. Nearest neighbour matching is used; we have added this information in the article as the description of the matching method was missing.

Clouds not detected at all by CPR (CloudSat) are not a problem as it simply means that we will have less data. That the CPR often misses the top portions could partly explain why results are not improving for NN-MetImage and NN-MetImage-NoCO2 (compared to NN-MERSI-2) when validating with CloudSat. We have added this discussion. The vertical resolution of CPR (CloudSat) is 0.5km this means that we should expect MAE higher than 250m, this information is also added.

**Changes in manuscript:**

- Matching described at page 6 line 4-5.
- Motivation for using CPR (CloudSat) included at page 8 line 20-26.
- Discussion of differences in result between CPR (CloudSat) and CALIOP added at page 12 line 11-16, page 13 line 6-11 and page 14 line 5-10.

**2.8 Referee comment:**

Pg.8, 26: The plots are distributions of the differences. Bias is the average of those differences. Please correct.

**Reply:**

We have correct that.

**Changes in manuscript:**

- page 35 line: 1

**2.9 Referee comment:**

Sec. 5 The discussion section is very thin. There is a paucity of what the results shown in the figures and table might mean. For example, what do the differences computed using two different references, CALIOP and CPR, tell us? All samples, except in polar regions are taken in midday or near midnight for Aqua. Could there be any diurnal impacts of training only with this dataset? What happens if the neighbouring pixel is turned off in the training? The conclusions state that that is an important input. Can its impact be quantified to support that conclusion?

**Reply:**

We thank Referee 1 for the suggestions and valuable comments that will help to improve the discussion.

The usage of two validation truths strengthens our results. The CloudSat results confirm that the improvements are not only due to that the neural networks have learnt to replicate errors of CALIOP. (For the argumentation let us pretend that CALIOP would always place clouds at 5km height if the surface pressure is 1000 hPa, a neural network could learn this but it would not really improve the accuracy of the retrieved cloud top height). Considering the large improvement it was not an alarming risk that the neural network was learning only to mimic CALIOP errors, but with the independent validation truth CloudSat this is confirmed. We have better motivated the inclusion of CloudSat in the paper.

**Changes in manuscript:**

- page 8 line: 20-26

What happens if the neighbouring pixels are not used is better described. We have discussed these results in more detail to support better the statement in the conclusion.

**Changes in manuscript:**

- page 9 line: 18-20
- The conclusion was moved and reformulated at page 17 line 16-18 (moved from page 16 line 22-23).

There might be diurnal impact not captured in the current dataset. However results are valid for Aqua which we trained for. Applying similar neural networks to other sensors with different filter functions and ECT will require additional work or validation not in the scope of this paper.

**2.10 Referee comment:**

Pg. 9, 22: It seems that using matches with Terra will not help much in the non-polar regions. Is this a realistic possibility given the orbital differences?

**Reply:**

As latitude is not used as a variable, data for higher satellite zenith angles included for Polar regions could help also in non-Polar regions. However it might be that the high latitude matches will not help the network the if varity of weather situations and cloud heights at high latitudes are too small. This must be tested. We have extend the discussion regarding adding Terra matches.

**Changes in manuscript:**

- page 15 line: 20-23

**2.11 Referee comment:**

Pg. 9, 30: This section is where the futher work on the sources of error (e.g., various cloud types) could be presented. It would help the discussion considerably.

**Reply:**

The Validation with CALIOP separated for different cloudtypes where included to answer the question on sources of errors from different cloud types. This was not possible to do with version 3 of CALIOP data as several of the classes of the feature classification are empty for CALIOP version 3 data. Therefore the validation was updated to use CALIOP version 4 data. As the validation with CALIOP are now done with the latest version, also the CPR (CloudSat) was updated to use the most recent version. The discussion and result section where merged into one section with several subsections. And the validation for different cloud types where included in section 4.5 *Validation with CALIOP separated for different cloudtypes*.

**Changes in manuscript:**

- Update of used version page 4 line 12-14, 18.
- A new Section 4.5 with results for different cloud types at page 14.
- Updates to the conclusion section at page 16 line 26-27 and page 17 line 5-6.
- Included the results for different cloud types in a new table at page 33.
- Figures, Tables and results concerned were updated to use the new versions.

**2.12 Referee comment:**

Sec. 6. More analysis in the discussion section would help flesh out this section.

**Reply:**

We have extended the Conclusion section, reflecting what was added to the Results and Discussion section.

**Changes in manuscript:**

- page 16 and 17 Section 5
- One scentence at page 17 line 29-20 were reformulated to be clearer.

**Reply to anonymous Referee 2 comments to** *Neural network cloud top pressure and height for MODIS**

Nina Håkansson et al.

**1** Overall quality of the discussion paper ("general comments"):**

**1.1 Referee comment:**

In the paper a novel retrieval of cloud top pressure and height using neural networks is presented. The presented retrieval technique is state of the art and an accurate technique. To account for different availabilities of channels on different satellites, a few modification of the neural network are investigated revealing the information content of the different channels. The new algorithms are compared to two reference algorithms, the CTTH algorithm of the NWCSAF PPS-v2014 and the MODIS collection 6 L2 height product. Additionally the algorithms is compared to CALIOP and CPR measurements. The quality of the algorithm is evaluated in terms of the mean absolute error (MAE). The improved quality of the results is impressive. From my point of view, I would request for at least another quality measure like standard deviation (similar to Tables 6 and 7). In overall, good work!

**Reply:**

We thank Referee 2 for this positive comment, and for the other valuable comments that will help us improve the paper further.

Regarding adding standard deviation, which was also requested by Referee 1, we chose the MAE as evaluation metric, over bias and standard deviation of differences (SDD), for many good reasons. However, some of the good reasons became clear to us first when we where faced with the request to include them in the article. Most important is that including bias and SDD of the error distribution intuitively gives the reader the mental picture of a Gaussian error distribution, centred at the bias. However we are dealing with skewed and even bimodal distributions as shown in Figure 2 and the mean is not at the centre of the distribution; i.e. the bias is not located at the peak of the error distribution.

The overall standard deviation is much affected by the largest errors. Some large errors are expected due to the differences between the passive and active sensors and the different FOV (field of view). Therefore we argue that the MAE is a better measure of variation of the error compared to SDD. The largest errors are of course also interesting, but when investigating these some care should be made to separate true errors from expected differences due to for example cloud edges in the FOV. We have included also the interquartile range (IQR) as it is more robust measures of variability less sensitive to outliers compared to SDD.

As the bias and SDD are traditionally used when evaluating cloud top height retrieval algorithms these are included along with the discussion of why they are not so useful or even misleading. To help the discussion also IQR, RMSE (root mean square error), mode, median and percentage of absoulte errors above 0.25, 0.5, 1 and 2 km were included.

**Changes in manuscript:**

- See reply to Referee 1 point 2.6.

**2 Individual scientific questions/issues ("specific comments")**

**2.1 Referee comment:**

p1 line 23: CTH might also be used in data assimilation of atmospheric motion vectors.

**Reply:**

We have added this to the text.

**Changes in manuscript:**

- page 2 line 8

**2.2 Referee comment:**

Introduction: A short description of the traditional technics to retrieve cloud top pressure and height could be added to the introduction. (or cite an overview paper like Hamann et al. "Remote sensing of cloud top pressure/height from SEVIRI: analysis of ten current retrieval algorithms." Atmospheric Measurement Techniques 7.9 (2014): 2839-2867.)

**Reply:**

We have added the suggested reference.

**Changes in manuscript:**

- page 2 line 12-13

**2.3 Referee comment:**

The introduction should motivate, why it is expected that using machine learning, in particular neural networks, could improve the expected results.

**Reply:**

Many CTH retrieval algorithms including MODIS-C6 and PPS-v2014 include some fitting of temperatures to NWP temperature profiles. This is most difficult in the case of inversions both as one temperature occurs at several pressure heights in the profile but also as the inversions are often not captured accurately enough in the NWP temperature profile. Many different techniques are used to deal with this, for example PPS-v2014 will place the cloud at the inversion height if the temperature is not more than 0.5 to 2K lower than the temperature at the inversion. MODIS-C6 has another approach using climatological lapse rates over sea for clouds likely to be low. These kinds of fitting techniques a statistical machine learning technique could probably do better. A motivation is included in the introduction.

**Changes in manuscript:**

- page 3 line 1-3

**2.4 Referee comment:**

Merge chapter 2.1.1 into chapter 3.2. (and skip the sentence (p3 line 10) "The MODIS Collection 6 cloud product were used as an independent. . .", you said that before).

**Reply:**

We have removed section 2.1.1 and moved the information from it to section 2.1 and removed the repeated information.

**Changes in manuscript:**

- page 3 line 28, 30, 31

**2.5 Referee comment:**

Chapter 2.2 Add a short sentence, why you chose the CALIOP 1km product and not also 5km or 10km product which are more sensitive to optically thin clouds.

**Reply:**

The 1km CALIOP product was selected because it has the resolution closest to the MODIS resolution. It is expected that the thinnest cloud seen by CALIOP lidar is invisible to the passive imagers, so it should not be a problem that the thinnest clouds are missing in the 1km data. However we have also done some tests using AVHRR-GAC data and CALIOP 5km (Version 4) resolution for training (this is outside the scope of this article). The first tests show that results improve if the thinnest (0.05 or 0.1 in optical depth) clouds are excluded from the training. If these networks (trained on AVHRR-GAC) are applied on the validation data (MODIS) of this article the MAEs of the retrievals are between 76hPa to 79hPa. We have added a sentence about why 1km data was chosen.

**Changes in manuscript:**

- page 4 line 12

**2.6 Referee comment:**

Chapter 2.4 add the version number of the ECMWF model and add product name and version number of the OSISAF data used in this study.

**Reply:**

We have added the versions and product names in section 2.4.

**Changes in manuscript:**

- page 4 line 24-26

**2.7 Referee comment:**

You might consider to add the PPS-v2014 and MODIS C6 algorithm to table 3 and 4.

**Reply:**

We have added them to Table 4, this will give the clear view of what channels are used for which method also for them. We suggest that they are not included in Table 3 as it describes *Network specific variables*. There was also an error in table 4, the NN-OPAQUE uses channel 12µm as described in Table 3 not channel 11µm. We have correct this as well and sorted the columns from lowest to highest wave length.

**Changes in manuscript:**

- page 25 Table 4

**2.8 Referee comment:**

Please make the order of algorithms in table 3, 4 and 5 consistent.

**Reply:**

We did this in the first revision, and can not find any remaning inconsistencies.

**2.9 Referee comment:**

p5 line 5: how often is a pressure lower than 70 hPa retrieved?

**Reply:**

It varies with each network from 0 up to 0.05%, we have added this to the text.

**Changes in manuscript:**

- page 6 line 9-10

**2.10 Referee comment:**

p5 line 10: Why did you choose this number of levels? Is it sufficient to use 6 levels to represent the boundary layer inversions or other small scale features?

**Reply:**

Five of the levels (surface, 950, 850, 700, 500) where already used in the PPS-v2014, so this was our starting point. We tested to use the troposphere pressure, but then the networks became very sensitive to the type of NWP-data used. Instead we added the 250hPa level to have one more high level. We did tested increasing the number of levels near the ground by adding levels at 800, 900 and 1000hPa, but the improvement was not large enough to motivate the extra computational time. One common problem for cloud height retrieval algorithms is that inversions are not represented accurately enough in the NWP data. As mentioned previously, MODIS-C6 instead uses climatological lapse-rates over sea to avoid this problem, other algorithms use sharpening techniques at the inversion. So it is not clear that more levels which would better represent the inversions in the NWP data would improve the neural network results, but this could be further investigated.

**2.11 Referee comment:**

p5 line 21: do you skip non cloudy pixels in the 5x5 pixel standard deviations?

**Reply:**

No, all pixels are included.

**2.12 Referee comment:**

p5 line 20: the  $B_3$ .7 has a solar component. Did you correct for this during day/night?

**Reply:**

No correction was made, that channel is used just like the others. The information that the solar component is not treated was added to the text.

**Changes in manuscript:**

- page 9 line 26-27

**2.13 Referee comment:**

p6 chapter 3.3.2: You chose to use specific days for training and others for validation. Given that you only use a limited number of days, wouldn't it be more to randomly select independent pixels from all available dates for training, validation, and testing to represent a larger variety of weather situations?

**Reply:**

Unfortunately all pixels in the dataset we have are not independent. A typical cloud is much larger than one pixel; there could be hundreds of pixels with almost identical data in the datasets. If we would randomly select independent pixels from all days for each dataset we would in practice use the same data all the time. This would cause the neural network to overtrain as the *during training validation data* would be the same as the *training data*. And in the last validation step results would be overly positive as also the *validation data* would be in practice the same as the *training data*.

Random sampling from all data has been suggested to us previously. We have therefore added the discussion to the article why it is not possible.

**Changes in manuscript:**

- page 3 line 25-27

**2.14 Referee comment:**

p 6 line 19: Did you test other configurations that 30/15 neurons in the first/second layer? If yes, how was the performance?

**Reply:**

We did one test with 20/15 this network (NN-AVHRR) was 1 hPa worse and one with 30/45/45 (NN-AVHRR) this was 2.5hPa better but also took 5 times as long time to retrieve pressure. This information have been included in the text.

**Changes in manuscript:**

- page 15 line 28-31

**2.15 Referee comment:**

Chapter 3.3.3: May batch size and momentum be changed during the training process?

**Reply:**

No they can't.

**2.16 Referee comment:**

p 7 line 27: consider to discuss the solar component of the 3.7 mue m channel. To my opinion this NN could perform better when corrected for that (e.g. adding the solar time as input variable).

**Reply:**

We considered adding the sun zenith angle as variable; however we can not decide how the neural network would use it. In the data we do not have all sun zenith angles present globally. It could be that the neural network would use the sun zenith angle to decide that during this time of day clouds of a particular height are most common. It does not have to be bad though and can be tested in future studies. The performance of NN-AVHRR1 could probably be improved it the solar component of 3.7 is treated explicitly. We have added discussion about the solar component of 3.7 µm.

**Changes in manuscript:**

- page 9 line 26-27

**2.17 Referee comment:**

p8 line 5: Maybe express it positively: All NN can reproduce a clear bi-modal pdf very similar to CALIPSO, the pdf of PPS-v2014 deviates from this shape . . .

**Reply:**

We have changed the formulation, thank you for the suggestion.

**Changes in manuscript:**

- page 10 line 3-7

**2.18 Referee comment:**

p8 line 7: It is written "for the best performing network". Did you train several networks for one channel configuration? If so, could you describe the number of trained networks in chapter 3.2.2, please?

**Reply:**

With the *the best performing networks* we meant that the NN-NWP, NN-OPAQUE, NN-BASIC and NN-BASIC-CIWV was excluded. We have clarified this in the text.

**Changes in manuscript:**

- page 10 line 22-23

**2.19 Referee comment:**

p 8, line 11: according to my table 6, the NN-MetImage is better than the NN-MetImage- NoCO2.

**Reply:**

Yes it does! However the NN-MetImage does not perform well for higher satellite zenith angles. Only networks that perform well for all satellite zenith angles are discussed in this sentence. We have added a sentence to make it clearer.

**Changes in manuscript:**

- page 12 line 21-22

**2.20 Referee comment:**

p 8, line 15: do you have an idea, why the MAE against CPR is larger than the MAE against CALIOP for NN-MetImage and NN-MetImage-NoCO2?

**Reply:**

This we think it partly because the NN-MetImage and NN-MetImage-NoCO2 have some skill in predicting very thin high clouds that are not detected by the CPR-Radar. We have added discussion about this.

**Changes in manuscript:**

- page 12 line 11-16

**2.21 Referee comment:**

p 9, line 9: could you please describe a bit more in detail the differences seen in Figure 7?

**Reply:**

We have described the differences in more detail.

**Changes in manuscript:**

- page 9 line 1-5 (moved from page 15 line 4-6 due to the reorganization of the Result section)

**2.22 Referee comment:**

Chapter 5 Discussion: Could you also comment on applying your NN technique on geostationary satellites? What would be the main differences/challenges?

**Reply:**

This technique should not be limited to polar orbiting satellites. As the instrument SEVIRI has the two most important channels at  $11\mu$ m and  $12\mu$ m it should be possible to apply the technique to SEVIRI data. More data (in terms of number of days) compared to MODIS may be needed to produce enough matches. As the SEVIRI resolution is coarser results might be degraded compared to MODIS. Matches of SEVIRI with CALIOP will occur at many different satellite zenith angles. This might make it possible to use the CO2 channel on SEVIRI to improve results without losing performance skill at high satellite zenith angles. We have commented on using the NN-CTTH technique for geostationary satellites in the paper.

**Changes in manuscript:**

- page 16 line 16-19

**A compact listing of purely technical corrections ("technical corrections": typing errors, etc.)**

**2.23 Referee comment:**

- p5 line 29 and thereafter: don't write CO2 with cursive letters
- consider to write NoCO2 (in MetImageNoCO2) not in cursive letters.

**Reply:**

We have kept the notation with subscript but without cursive letters. All CO2 are updated to be written without cursive letters.

**2.24 Referee comment:**

Figure 1 (p21): consider to have the figures in the same order as the algorithms are mentioned in table 3, 4, and 5.

**Reply:**

This is a reasonable request. However it is also nice to have the two AVHRR based algorithms next to each other so they can be compared. Also this order makes the two networks performing bad at high satellite zenith angles appear on the last row. And changing the order increases the risk to mix them up in later references. If someone refers to the bad satellite zenith angle behaviour in Figure 2 (h) in the discussion paper, and an interested reader by accident finds the final revised paper (assuming

there will be one) and finds the result for NN-AVHRR1 in that sub figure that is not good. As there are also good reasons to keep the current order of sub figures we argue that ther order should not be changed.

**2.25 Referee comment:**

- p 8, line 34 and following: avoid the abbreviation NN-CTTH, e.g. change: that the NN-CTTH all have -> that all NN retrievals have . . .
- avoid NN-CTTH abbreviation (which one do you mean? all NN retrievals or NN-MetImage or another one?)
- p 10, line 12: avoid NN-CTTH -> specify which retrieval you referring to

**Reply:**

We have kept the NN-CTTH as the name for the neural network method in the paper. We have present it as the name and avoided using it where it might be confusing.

**Changes in manuscript:**

- page 8 line 29 Reformulated from NN-CTTH all have. Note that the lines where moved from page 13 line 35.
- page 16 line 24 Clearified that NN-CTTH means all networks in this sentence.
- page 16 line 33 Reformulated to not use NN-CTTH abbreviation.

**2.26 Referee comment and changes in manuscript:**

- C: Please check consistent spelling of NWC SAF (e.g. p2 line 11) and NWCSAF (e.g. p1 line 19)
- R: Changed to use NWC SAF page 1 line 7 and page 17 line 23.
- C: Please check consistent spelling of PPS-2018 (e.g. p1 line 19) and PPS-v2014 (e.g. p2 line 23)
- R: Sentence removed page 2 line 2.
- C: Please check the space between numbers and units and the typeset of the units.
- R: Done
- C: Try to reduce number of paragraphs in the abstract, e.g. p1 line 8 is a one sentence paragraph.
- R: The one sentence paragraph was merged with the following paragraph. However the new paragraph about the statistical measures means that there are still 5 paragraphs in the abstract.

- C: please check capital letters, e.g. Neural network (p3 line 26), neural network (p2 line 14) or Neural Network.
- R: Updated to use neural network without capital letters at page 4 line 23.
- C: p3 line 22: (change . to ,) . . . of the networks, see Table 1 for selected Dates
- R: Changed at page 4 line 19.
- C: Move p4 line 9-14 to line 6.
- R: Moved page 5 line 15-18 to line 7-10.
- C: p4 line 27: introduce abbreviation GDAS (as written in line 30)
- R: Abbrevation introduced page 5 line 31-32.
- C: p4 line 28: add "the": . . . and the PFAAST radiative transfer model. . .
- R: Added "the" at page 5 line 32.
- C: p5 line 3 add: The "uppermost cloud" top layer. . .
- R: Added at page 6 line 6.
- C: p5 line 11: reformulate "much of what"
- R: Reformulated page 6 line 15-16.
- C: p5 line 19: introduce physical unit "B" (in the lines before)
- R: Unit introduced at page 6 line 15.
- C: p5 line 23: B\_11 "for" neighboring pixels -> B\_11 "of the" neighboring Pixels
- R: Changed at page 6 line 28.
- C: p5 line 25: avoid brackets
- R: Reformulated without brackets at page 6 line 29-30.
- C: p 15 line 7, add "and": BT for water vapour channels at 6.7 "and" 7.3 mue m
- R: Added or at page 23 line 7.
- C: p15 line 9/10/11, remove "." at the end of first entry, e.g. BT differences ""
- R: Removed the "." at page 23 line 9-11.

- C: p5 line 29 and thereafter: don't write CO2 with cursive letters
- R: Changed to use non-cursive letters.
- C: p 6 line 19: hidden layer "for" the neural network -> hidden layer "of" the neural network
- R: Changed at page 7 line 25.
- C: p 6 line 17-32: reduce the number of paragraphs. Don't create one sentence para- graphs.
- R: Paragraph 2 and 3 of Section 3.3.3 were merged.
- C: p 7 line 18: write Ciwv in cursive letters
- R: Changed at page 9 line 14.
- C: p 7 last line: N-VIIRS -> NN-VIIRS
- R: Corrected at page 9 line 33.
- C: Figure 7 (p27): Could you please add a color scale instead of describing it with words.
- R: Color scale added and the description with words removed at page 40.
- C: p 9, line 11 and thereafter: cloud heights -> cloud "top" heights,
- R: Done
- C: p 10, line 15: The NN CTTH retrievals all have better results for low, medium and high clouds . . . (Clouds don't show results. . .)
- R: Reformulated at page 16 line 26.

**2.27 Additional changes:**

Three additional scentences were slightly reformulated to be clearer:

- page 7 line 18
- page 9 line 12
- page 15 line 1

[revised manuscript text omitted]
.algorithm.com/algorithm/algorithm/algorithm/algorithm/algorithm/algorithm/algorithm/algorithm/algorithm/algorithm/algorithm/algorithm/algorithm/algorithm/algorithm/algorithm/algorithm/algorithm/algorithm/algorithm/algorithm/algorithm/algorithm/algorithm/algorithm/algorithm/algorithm/algorithm/algorithm/algorithm/algorithm/algorithm/algorithm/algorithm/algorithm/algorithm/algorithm/algorithm/algorithm/algorithm/algorithm/algorithm/algorithm/algorithm/algorithm/algorithm/algorithm/algorithm/algorithm/algorithm/algorithm/algorithm/algorithm/algorithm/algorithm/algorithm/algorithm/algorithm/algorithm/algorithm/algorithm/algorithm/algorithm/algorithm/algorithm/algorithm/algorithm/algorithm/algorithm/algorithm/algorithm/algorithm/algorithm/algorithm/algorithm/algorithm/algorithm/algorithm/algorithm/algorithm/algorithm/algorithm/algorithm/algorithm/algorithm/algorithm/algorithm/algorithm/algorithm/algorithm/algorithm/algorithm/algorithm/algorithm/algorithm/algorithm/algorithm/algorithm/algorithm/algorithm/algorithm/algorithm/algorithm/algorithm/algorithm/algorithm/algorithm/algorithm/algorithm/algorithm/algorithm/algorithm/algorithm/algorithm/algorithm/algorithm/algorithm/algorithm/algorithm/algorithm/algorithm/algorithm/algorithm/algorithm/algorithm/algorithm/algorithm/algorithm/algorithm/algorithm/algorithm/algorithm/algorithm/algorithm/algorithm/algorithm/algorithm/algorithm //www.nwcsaf.org/AemetWebContents/ScientificDocumentation/Documentation/PPS/v2014/NWC-CDOP2-PPS-SMHI-SCI-ATBD-3

35 v1\_0.pdf, 2015.

[revised manuscript text omitted]

 Table 4. Description of the imager channels used for the different networks algorithms. For MODIS-C6 channels used indirectly, to determine if CO2-slicing should be applied, are noted with brackets.

|                        | Imager channel:            | B11-B3.7 | $B_{12} - B_{6.7}$ | B7.3 | $B_{8.5}$   | $B_{7.3}$ $B_{11}$ | $B_{6.7}$ $B_{12}$ | $B_{13.3}$           | $B_{3.7} \xrightarrow{B_{13.6}}$ | $\underline{B_{13.9}}$ |
|------------------------|----------------------------|----------|--------------------|-------------|-------------|--------------------|--------------------|----------------------|----------------------------------|------------------------|
| Network name           |                            |          |                    |             |             |                    |                    |                      |                                  |                        |
| PPS-v2014              |                            |          |                    |             |             | ×~                 | ×~                 |                      |                                  |                        |
| MODIS-C6               |                            |          |                    | (x)  | ( x) | ×                  | (x)         | $\stackrel{X}{\sim}$ | $\stackrel{\rm X}{\sim}$         | $\stackrel{X}{\sim}$   |
| NN-NWP                 |                            |          |                    |             |             |                    |                    |                      |                                  |                        |
| NN-OPAQUE              |                            |          |                    |             |             |                    | х                  |                      |                                  |                        |
| NN-BASIC               |                            |          |                    |             |             | х                  | х                  |                      |                                  |                        |
| NN-BASIC-CIWV          |                            |          |                    |             |             | х                  | х                  |                      |                                  |                        |
| NN-AVHRR               |                            |          |                    |             |             | х                  | х                  |                      |                                  |                        |
| NN-VIIRS               |                            |          |                    |             | х           | х                  | х                  |                      |                                  |                        |
| NN-MERSI-2             |                            |          |                    | х           | х           | х                  | х                  |                      |                                  |                        |
| NN-MetImage-NoCO2 NN-N | MetImage-NoCO 2 |          | Х                  | х           | х           | х                  | х                  |                      |                                  |                        |
| NN-MetImage            |                            |          | Х                  | х           | х           | х                  | х                  | х                    |                                  |                        |
| NN-AVHRR1              |                            | х        |                    |             |             | х                  |                    |                      |                                  |                        |

**Table 5.** Mean absolute error (MAE) for different algorithms compared to CALIOP top layer pressure. The final validation dataset (see Table 1), containing 1796428-1832432 pixels (45%-% high, 39%-% low and 16%-% medium level clouds) is used. Pixels with valid pressure for PPS-v2014, MODIS-C6, and CALIOP are considered. The low, medium and high classes are from CALIOP feature classification flag.

|                                     |                                 | MAE                          | [hPa]                        |                                 |
|-------------------------------------|---------------------------------|------------------------------|------------------------------|---------------------------------|
|                                     | all                             | low                          | medium                       | high                            |
| PPS-v2014                           | <del>122.9</del> 122.6          | <del>79.4</del> 80.2         | <del>88.6</del> 88.0         | <del>173.5</del> 172.9          |
| MODIS-C6                            | <del>124.3123.9</del>    | <del>90.4</del> 90.7         | <del>140.0139.8</del> | <del>148.4147.3</del>    |
| NN-NWP                              | <del>191.6</del> - 191.7 | <del>140.8141.7</del> | <del>110.5</del> 110.3       | <del>266.0-265.8</del>          |
| NN-OPAQUE                           | <del>113.3-</del> 113.2  | <del>81.382.1</del>   | <del>105.1</del> 105.0       | <del>144.5</del> - 143.8 |
| NN-BASIC                            | 93.9                            | <del>66.767.7</del>   | 92.8                         | <del>118.3-117.6</del>          |
| NN-BASIC-CIWV                       | 92.1                            | <del>66.4</del> 67.5  | <del>91.4</del> 91.3         | <del>115.0</del> - 114.2 |
| NN-AVHRR                            | <del>72.2</del> -7 2.4   | <del>54.1</del> 55.4  | <del>67.4</del> 67.6         | <del>89.9</del> 89.2            |
| NN-VIIRS                            | <del>65.7-65.9</del>            | <del>49.1</del> 50.5  | <del>59.2</del> 59.3         | <del>82.7-81.9</del>            |
| NN-MERSI-2-NN-MERSI2                | <del>61.2-61.4</del>            | <del>46.7</del> 48.2  | <del>52.052.1</del>          | <del>77.3</del> -7 6.6   |
| NN-MetImage-NoCO2-NN-MetImage-NoCO2 | <del>60.0-60.3</del>            | 45.5 47.1             | <del>54.3</del> 54.5         | <del>74.8</del> -74.1           |
| NN-MetImage                         | <del>53.6-</del> 54.2           | <del>42.7</del> 44.5  | <del>51.3</del> 51.6         | <del>64.1-63.8</del>            |
| NN-AVHRR1                           | 76.1                            | <del>53.6</del> 54.7         | <del>70.069.9</del>          | <del>98.1</del> -97.3           |

**Table 6.** Mean absolute error (MAE) in meters for different algorithms compared to CALIOP top layer altitude. The final validation dataset (see Table 1), containing 1793142 pixels (45% high, 39% low and 16% medium level clouds), where all algorithms had a cloud height is used. The low, medium and high classes are from CALIOP feature classification flag. A small amount 0.2% of the pixels were excluded because of missing height or pressure below 70hPa for any of the algorithms.

| MAE                           |      |     |        |      |
|-------------------------------|------|-----|--------|------|
|                               | all  | low | medium | high |
| PPS-v2014                     | 2087 | 837 | 1124   | 3542 |
| MODIS-C6                      | 1917 | 944 | 1759   | 2833 |
| NN-AVHRR                      | 1290 | 567 | 962    | 2049 |
| NN-VIIRS                      | 1177 | 514 | 828    | 1891 |
| NN-MERSI-2                    | 1110 | 488 | 727    | 1797 |
| NN-MetImage-NoCO 2 | 1081 | 478 | 757    | 1732 |
| NN-Metimage                   | 964  | 451 | 707    | 1510 |
| NN-AVHRR1                     | 1375 | 560 | 978    | 2239 |

**Table 7.** Mean absolute error (MAE) in meters for different algorithms compared to CPR (CloudSat) Height. The final validation dataset (see Table 1), containing 1121199 pixels (53% high, 27% low and 21% medium level clouds) is used. The low, medium and high classes are derived comparing the CloudSat height to the NWP height at 440hPa and 680hPa. A cloudy threshold of 30% is used for CloudSat.

| MA                            |      |      |        |      |
|-------------------------------|------|------|--------|------|
|                               | all  | low  | medium | high |
| PPS-v2014                     | 1761 | 977  | 1365   | 2315 |
| MODIS-C6                      | 1711 | 1206 | 1912   | 1888 |
| NN-AVHRR                      | 1278 | 771  | 1218   | 1559 |
| NN-VIIRS                      | 1223 | 766  | 1144   | 1486 |
| NN-MERSI-2                    | 1135 | 748  | 1061   | 1362 |
| NN-MetImage-NoCO 2 | 1161 | 768  | 1095   | 1386 |
| NN-Metimage                   | 1186 | 802  | 1119   | 1407 |
| NN-AVHRR1                     | 1297 | 858  | 1225   | 1548 |

**Table 8.** Statistic measures for the error distributions for all clouds. For all measures except skewness it is the case that values closer to zero are better. The statistics are calculated for 1198599 matches for CPR (CloudSat) and 1803335 matches for CALIOP. A small amount 0.2% of the matches were excluded because of missing height or pressure below 70 hPa for any of the algorithms. PEX describes percentage of absolute errors above X km, see Equation 1.

|                               | MAE          | IQR         | RMSE         | $\underbrace{SD^1}$ | $\underbrace{PE_{0.25}}_{\sim}$ | PE 0.5 | $\underbrace{PE_{1}}_{\sim}$ | $\underbrace{PE_{2\sim}}$ | median       | mode         | $\underline{bias}^1$ | skew         |  |
|-------------------------------|--------------|-------------|--------------|---------------------|---------------------------------|-------------------|------------------------------|---------------------------|--------------|--------------|----------------------|--------------|--|
|                               | [ m ] | [ m] | [ m ] | [ m]         | [%]                             | [%]               | [%]                          | [%]                       | [ m ] | [ m ] | [ m ]         |              |  |
| CALIOP all clouds             |              |             |              |                     |                                 |                   |                              |                           |              |              |                      |              |  |
| PPS-v2014              | 2095         | 2832        | 3188         | 2832                | .82                      | 69         | .54                          | .29                       | -639         | -118         | -1465                | - 1.0 |  |
| MODIS-C6                      | 1923         | 2177        | 3105         | 2883                | .85                      | 72                | .51                   | 23                        | -612         | -262         | -1153                | - 1.5 |  |
| NN-AVHRR                      | 1300         | 1326        | 2234         | 2197         | .73                             | 5 6        | .36                   | .14                       | 50    | _106_        | -405                 | - 1.8 |  |
| NN-VIIRS                      | 1187         | 1189        | 2114         | 2074                | .71                      | 5 2        | 33                           | 12                        | 28           | _100_        | -410                 | - 1.9 |  |
| NN-MERSI-2                    | 1120         | 1107        | 2039         | 1996         | .69                      | 5 0        | 30                    | 11                        | -2           | .73          | -420                 | -2.0         |  |
| NN-MetImage-NoCO 2 | 1091         | 1040        | 2009         | 1966         | .68                      | 48                | .29                   | 11                        | -49          | .44          | -416                 | -2.0         |  |
| NN-MetImage                   | .979  | 909  | 1840         | 1817                |                                 | 46         | .26                   | .9                        | -17          | 15           | -294                 | - 1.9 |  |
| NN-AVHRR1                     | 1383         | 1547        | 2354         | 2281                |                                 | 5 8        | 38                           | .16                | -42          | .50          | -584                 | - 1.8 |  |
|                               |              |             |              | CP                  | R (CloudSa                      | t) all clou       | ıds                          |                           |              |              |                      |              |  |
| PPS-v2014              | 1744         | 2255        | 2432         | 2160                | 87                       | 74                | .56                          | .24                       | -833         | -426         | -1118                | -0.1         |  |
| MODIS-C6                      | 1692         | 1928 | 2607         | 2533                |                                 | 70                | 48                    | 20                 | -375         | -259         | -614                 | - 0.1 |  |
| NN-AVHRR                      | 1262         | 1473 | 1928         | 1923         |                                 | 61                | 41                    | .14                       | 88           | -141         | 143                  | 0.2          |  |
| NN-VIIRS                      | 1207         | 1368        | 1901         | 1896                |                                 | 58                | .38                   | 13                        | 69    | -146         | 137                  | 0.5          |  |
| NN-MERSI-2                    | 1120         | 1275        | 1793         | 1788                |                                 | 5 6        | .35                          | 11                        | 40    | -201         | 136                  | 0.5          |  |
| NN-MetImage-NoCO 2 | 1146         | 1315        | 1834         | 1828                |                                 | 57                | 35                           | 12                        | 9_           | -218         | 147                  | 0.7          |  |
| NN-MetImage                   | 1170         | 1421        | 1865         | 1843                |                                 | 58                | 37                           | 11                        | 84           | -243         | 285                  | 0.9   |  |
| NN-AVHRR1                     | 1281         | 1523        | 1953         | 1953         | .79                             | 63                | 41                           | .14                       | 30           | -128         | -14                  | 0.0          |  |

**Table 9.** Statistic measures for the error distributions for low level clouds. For all measures except skewness it is the case that values closer to zero are better. The statistics are calculated for 328015 matches for CPR (CloudSat) and 709434 matches for CALIOP. The low class comes from CALIOP feature classification flag (class 0, 1, 2 and 3) and for CPR (CloudSat) it is the pixels with heights lower or exactly at the NWP height at 680 hPa. PEX describes percentage of absolute errors above X km, see Equation 1.

|                               | MAE          | IQR         | RMSE         | $\underbrace{SD}^1$ | PE 0.25~ | PE 0.5 | $\underbrace{PE_{1}}_{\sim}$            | $\underline{PE}_{2\sim}$ | median       | mode         | $\underbrace{\text{bias}}^1$ | skew |
|-------------------------------|--------------|-------------|--------------|---------------------|---------------------|-------------------|-----------------------------------------|--------------------------|--------------|--------------|------------------------------|------|
|                               | [ m ] | [ m] | [ m ] | [ m]         | [%]                 | [%]               | [%]                                     | [%]                      | [ m ] | [ m ] | [ m ]                 |      |
|                               |              |             |              | Lov                 | v level clou        | ds CALI           | OP                                      |                          |              |              |                              |      |
| PPS-v2014              |              | 1035        | 1469         | 1436                | .68          | 47                | .27                                     | 5                        | -46          | -117         | 312                          | 3.0  |
| MODIS-C6                      | .952         | 1230        | 1576         | 1561                |                     | 58                | .29                              | 6                 | -17          | -150         | 219                          | 2.9  |
| NN-AVHRR                      | .586         | 584  | 1121         | 1027                | .56                 | 31                | .14                                     | 3                        | 215          | _101  | 449                   | 4.0  |
| NN-VIIRS                      | .533         | 515  | 1080         | 1006                | .52          | 27                | 11                                      | 3                        | 182          | _126  | 391                   | 4.8  |
| NN-MERSI-2                    | .509         | 490  | 1063         | 998          | 49           | 25                | 10                                      | 3                        | 159          | .86   | 365                          | 4.8  |
| NN-MetImage-NoCO 2 | 499   | 504  | 1068         | 1024                | 48                  | 24                | 10                                      | 3                        | 98    | 40    | 303                          | 4.9  |
| NN-MetImage                   | .476  | 450  | 1103         | 1069                | 45                  | 21                | 8~~ | 3                        | 74           | .14          | 271                          | 5.4  |
| NN-AVHRR1                     | 574          | 646  | 1045         | 969          | .58          | 33                | 13                                      | 3                        | 197   | 49    | 391                          | 3.8  |
|                               |              |             |              | Low le              | vel clouds (        | CPR (Clo          | udSat)                                  |                          |              |              |                              |      |
| PPS-v2014                     | 949          | 1197        | 1571         | 1556                |                     | 56                | 29                               | 5                        | -173         | -413         | 211                          | 2.8  |
| MODIS-C6                      | 1192         | 1335        | 2145         | 2097         | .79          | 60                | 33                                      | 9                 | 46    | -110         | 450                          | 2.9  |
| NN-AVHRR                      | .743         | 685  | 1595         | 1532                | .56                 | 31                | .16                                     | 6                 | 16           | -132         | 443                          | 3.8  |
| NN-VIIRS                      | .739  | 637  | 1690         | 1633         | .55                 | 30                | .15                                     | 6                 | -6           | -139         | 432                          | 4.2  |
| NN-MERSI-2                    | .721         | 605  | 1652         | 1602                | .55                 | 28                | .14                                     | 6                 | -31          | -181         | 403                          | 4.1  |
| NN-MetImage-NoCO2             | .742         | 608  | 1670         | 1637         | .60          | 31                | 13                                      | 6                 | -105         | -255         | 328                          | 4.2  |
| NN-MetImage                   |              | 578  | 1813         | 1775                | .58          | 30                | 13                                      | 6                 | -102         | -217         | 369                          | 4.1  |
| NN-AVHRR1                     | .827         | 852  | 1676         | 1602                | .64                 | 38                | 18                                      | .7                       | 48           | -198         | 491                   | 3.6  |

**Table 10.** Statistic measures for the error distributions for medium level clouds. For all measures except skewness it is the case that values closer to zero are better. The statistics are calculted for 244885 matches for CPR (CloudSat) and 295186 matches for CALIOP. The high class comes from CALIOP feature classification flag (class 4 and 5) and for CPR (CloudSat) it is the pixels with heights between the NWP height at 440 hPa and 680 hPa. PEX describes percentage of absolute errors above X km, see Equation 1.

|                          | MAE                        | IQR         | RMSE         | $\underbrace{SD}^1$ | PE 0.25 | PE 0.5 | $\underline{PE}_{1}$ | $\underline{PE}_{2\sim}$ | median       | mode         | bias 1 | skew       |  |  |
|--------------------------|----------------------------|-------------|--------------|---------------------|--------------------|-------------------|----------------------|--------------------------|--------------|--------------|--------------------------|------------|--|--|
|                          | [ m ]               | [ m] | [ m ] | [ m]         | [%]                | [%]               | [%]                  | [%]                      | [ m ] | [ m ] | [ m ]             |            |  |  |
|                          | Medium level clouds CALIOP |             |              |                     |                    |                   |                      |                          |              |              |                          |            |  |  |
| PPS-v2014         | 1121                       | 1600        | 1651         | .1614        |                    | 59         | 37                   | 12                       | -68          | 124          | -348                     | 0.2        |  |  |
| MODIS-C6                 | 1759                       | 2590 | 2304         | 2192         |                    | 7 6        | .60           | .27                      | -654         | _205_        | -708                     | 0.6 |  |  |
| NN-AVHRR                 | .969                | 1243        | 1394         | 1339         |                    | 59         | .34                  | 7                        | 304          | .273         | 387                      | 0.8        |  |  |
| NN-VIIRS          | .832                       | 1048        | 1227         | 1206                | .74                | 53                | 28                   | 5                        | 186          | .23          | 223               | 0.7        |  |  |
| NN-MERSI-2               | .731                       | 935  | 1102         | 1093         | .70         | 47                | 23                   | 4~~~                     | 83           | .16          | 144                      | 0.9 |  |  |
| NN-MetImage-NoCO2 | .762                | 984  | 1148         | 1145                | .71                | 49                | .24                  | 4~~~                     | 28           | -1           | 86                | 1.1        |  |  |
| NN-MetImage              | .714                       | 905  | 1091         | 1090         | .69         | 46         | 22                   | 3~~~                     | 4            | - 63  | 36                       | 1.1        |  |  |
| NN-AVHRR1                | .980                | 1330        | 1381         | 1364                |             | 61                | 35                   | 7                        | 187          | .176         | 213                      | 0.5        |  |  |
|                          |                            |             | Ν            | Aedium              | level cloud        | s CPR (C          | loudSat)             |                          |              |              |                          |            |  |  |
| PPS-v2014         | 1364                       | 1978 | 1927         | 1858                | .82                | 66         | 46            | .18                      | -300         | .53          | -512                     | 0.5        |  |  |
| MODIS-C6                 | 1909                | 2698        | 2532         | 2475         | .88                | 7 8        | .62           | 30                | -597         | .69   | -534                     | 0.9        |  |  |
| NN-AVHRR                 | 1215                       | 1541        | 1817         | 1770                | .81                | 64                | 40            | 12                       | 209          | -113         | 409               | 1.2        |  |  |
| NN-VIIRS          | 1139                       | 1325        | 1788         | 1760                |                    | 59         | .36           | 11                       | 114          | -81          | 310                      | 1.5        |  |  |
| NN-MERSI-2               | 1059                       | 1203        | 1706         | 1686                |                    | 55                | 32                   | 10                | 15           | -150         | 264                      | 1.7        |  |  |
| NN-MetImage-NoCO2        | 1091                       | 1259        | 1752         | 1740                |                    | 57                | 33                   | 10                | -44          | -154         | 205                      | 1.8        |  |  |
| NN-MetImage              | 1113                       | 1217        | 1832         | 1818         | .75                | 5 6        | 33                   | 11                       | -45          | -174         | 225                      | 1.9        |  |  |
| NN-AVHRR1                | 1221                       | 1591 | 1776         | 1751         | .81         | 65                | 41                   | 13                       | 146          | -25          | 301                      | 1.0        |  |  |

**Table 11.** Statistic measures for the error distributions for high level clouds. For all measures except skewness it is the case that values closer to zero are better. The statistics are calculated for 625699 matches for CPR (CloudSat) and 798715 matches for CALIOP. The high class comes from CALIOP feature classification flag (class 6 and 7) and for CPR (CloudSat) it is the pixels with heights higher or exactly at the NWP height at 440 hPa. PEX describes percentage of absolute errors above X km, see Equation 1.

|                               | MAE                      | IQR         | RMSE         | $\underbrace{SD}^1$ | PE0.25 | PE0.5 | $\underline{PE}_{1}$ | $\underline{PE}_{2\sim}$ | median       | mode         | bias 1 | skew         |  |  |  |
|-------------------------------|--------------------------|-------------|--------------|---------------------|--------------------------|--------------|----------------------|--------------------------|--------------|--------------|--------------------------|--------------|--|--|--|
|                               | [ m ]             | [ m] | [ m ] | [ m]         | [%]                      | [%]          | [%]                  | [%]                      | [ m ] | [ m ] | [ m ]             |              |  |  |  |
|                               | High level clouds CALIOP |             |              |                     |                          |              |                      |                          |              |              |                          |              |  |  |  |
| PPS-v2014              | 3564                     | 3367        | 4475         | 2842                | 96                | 92    | .84                  | .57                      | -2918        | -1897        | -3456                    | -0.9         |  |  |  |
| MODIS-C6                      | 2846                     | 3095 | 4196         | 3342                | .92                      | 84           | .68           | .36               | -1586        | -917  | -2537                    | - 1.5 |  |  |  |
| NN-AVHRR                      | 2057                     | 2775        | 3072         | 2704                |                          | 7 6   | .57                  | .27                      | -799  | -130         | -1457                    | - 1.4 |  |  |  |
| NN-VIIRS                      | 1899                     | 2459        | 2916         | 2581                | .86               | 74           | .53                  | .23                      | -716         | -18          | -1356                    | - 1.6 |  |  |  |
| NN-MERSI-2                    | 1807                     | 2258        | 2818         | 2486                | .85                      | 72           | .51           | 21                       | -705         | -192         | -1326                    | - 1.7 |  |  |  |
| NN-MetImage-NoCO 2 | 1739                     | 2134        | 2760         | 2464                |                          | 70           | 48            | _20               | -606         | -248         | -1242                    | - 1.8 |  |  |  |
| NN-MetImage                   | 1524                     | 1906 | 2476         | 2298                | 83                       | 67           | 44            | .16               | -360         | -83          | -920              | -2.0         |  |  |  |
| NN-AVHRR1                     | 22 50             | 2913 | 3292         | 2791         | 89                | 7 9   | 61            | 30                | -1099        | -475         | -1746                    | - 1.3 |  |  |  |
|                               |                          |             |              | High le             | evel clouds              | CPR (Clo     | udSat)               |                          |              |              |                          |              |  |  |  |
| PPS-v2014              | 2309                     | 2384        | 2930         | 2092                | 93                | 87           | .74                  | .36               | -1789        | -1428        | -2052                    | -0.5         |  |  |  |
| MODIS-C6                      | 1869                     | 2142        | 2845         | 2578         | .86               | 73           | .51           | .22                      | -614         | -506         | -1203                    | - 1.2 |  |  |  |
| NN-AVHRR                      | 1553                     | 2244        | 2121         | 2118                |                          | 75           | .54                  | .19               | 143          | 348          | -117                     | - 0.6 |  |  |  |
| NN-VIIRS                      | 1479                     | 2095        | 2043         | 2041                | .86               | 73           | .52                  | .17                      | 168          | 332          | -85                      | -0.7         |  |  |  |
| NN-MERSI-2                    | 1353                     | 1876        | 1894         | 1893         | .85                      | 71           | 48            | .14                      | 177          | 326          | -54                      | - 0.9 |  |  |  |
| NN-MetImage-NoCO 2 | 1379                     | 1843        | 1944         | 1944         | 85                | 71           | 48            | .15                      | 219          | 292          | 29                       | -0.7         |  |  |  |
| NN-MetImage                   | 1399                     | 1871 | 1904         | 1885                |                          | 74           | .52                  | .14                      | 463          | 511          | 265                      | - 0.8 |  |  |  |
| NN-AVHRR1                     | 1542                     | 2275 | 2145         | 2107                |                          | 74           | .53                  | .19               | -67          | 281          | -403                     | - 0.8 |  |  |  |

**Table 12.** Mean absolute error (MAE) and median in meters for different algorithms compared to CALIOP top layer altitude. The final validation dataset (see Table 1), containing 1803335 pixels (5 % low overcast (transparent), 12 % low overcast opaque, 19 % transition stratocumulus, 2 % low, broken cumulus, 7 % altocumulus (transparent), 8 % altostratus (opaque), 30 % cirrus (transparent) and 14 % deep convective (opaque)), where all algorithms had a cloud top height is used. The cloud types are from CALIOP feature classification.  $PE_{0.5}$  describes percentage of absolute errors above 0.5 km.

|                               | ~                 | rent       |            |            | 1115         | is rent    |            |            |
|-------------------------------|-----------------------|------------|------------|------------|--------------|------------|------------|------------|
|                               |                       | trans      | Par opar   | he rocur   | nu unulu     | s (transf  | ia salle   | arent      |
|                               |                       | ercast.    | ercastr .  | on stra    | oken cr      | HIHS C     | atus (or   | ranspi     |
|                               | 100% 02               | 1004 02    | transit    | 100%       | allociti     | allost     | innis      | deep       |
|                               | MAE [m]               |            |            |            |              |            |            |            |
| PPS-v2014              | 709            | 637 | 886        | 1695       | 1609         | 699 | 4343       | 1901       |
| MODIS-C6                      | 903            | 1028       | 901 | 1058       | 2343         | 1254       | 3567       | 1308       |
| NN-AVHRR                      | 519            | 442 | 627 | 1027       | 1134         | 825 | 2608       | 883        |
| NN-VIIRS                      | 454                   | 407 | 571        | 938 | 1011         | 678 | 2398       | 833        |
| NN-MERSI-2                    | 408            | 381 | 550 | 946 | 900   | 584 | 2283       | 791 |
| NN-MetImage-NoCO2             | 395            | 372 | 541 | 929 | 929   | 617 | 2210       | 734        |
| NN-MetImage                   | 365                   | 364        | 509 | 912 | 885          | 565 | 1905       | 711        |
| NN-AVHRR1                     | 516            | 448 | 617        | 911 | 1156         | 827 | 2847       | 977 |
|                               | median [m]            |            |            |            |              |            |            |            |
| PPS-v2014                     | -183                  | 50         | -90        | 220        | -633         | 63         | -3835      | -1716      |
| MODIS-C6                      | -91                   | 331        | -138       | -477       | -1953        | 85  | -2243      | -912       |
| NN-AVHRR                      | 223                   | 160        | 241        | 380        | 109          | 410 | -1605      | 71         |
| NN-VIIRS                      | 185                   | 143        | 201        | 315        | .7           | 279 | -1360      | 46         |
| NN-MERSI-2                    | 160                   | 116        | 177        | 313        | -34          | 145        | -1268      | -35        |
| NN-MetImage-NoCO 2 | 110                   | 70  | 102        | 226        | -138         | 119        | -1133      | -19        |
| NN-MetImage            | 53                    | 46  | 86         | 214        | - 163 | 87  | -787       | 125        |
| NN-AVHRR1                     | 188                   | 140        | 232        | 313        | -180         | 380        | -1895      | -82        |
|                               | PE 0.5 [%] |            |            |            |              |            |            |            |
| PPS-v2014              | 46                    | 38         | 49         | 67  | 76           | 44         | 95         | 87         |
| MODIS-C6                      | 58                    | 59  | 56         | 63  | 89           | 64  | 87         | 76         |
| NN-AVHRR                      | 33                    | 23         | 34         | 47  | 67           | 53  | 83         | 60         |
| NN-VIIRS                      | 27                    | 19         | 30         | 43         | 63           | 44  | 81         | 58         |
| NN-MERSI-2                    | 2 4            | 17         | 28  | 42  | 58           | 37         | 7 9 | 57         |
| NN-MetImage-NoCO 2 | 22                    | 16  | 27         | 40  | 59           | 40  | 78         | 53         |
| NN-MetImage                   | 20             | 15         | 23         | 37         | 58           | 36  | 74         | 52  |
| NN-AVHRR1                     | 34                    | 24  | 37         | 45  | 68    | 54  | 86         | 64         |